# Spectral-Progressive Thought Flow for Lightweight Multimodal Reasoning

Yixian Shen [1 2]  Zhiheng Yang [1]  Qi Bi [1]  Changshuo Wang [3]  Shuai Wang [1]  Jia-Hong Huang [1 4]  George Floros [5]
Prayag Tiwari [6]  Anuj Pathania [1]

## Abstract

Multimodal spatial reasoning often relies on long chains of intermediate textual and visual thoughts, where accumulating visual tokens and dense cross-modal attention incur substantial computation and memory overhead. To address this challenge, we propose Spectral-Progressive Thought Flow (*SpecFlow*), a *novel* lightweight multimodal spatial reasoning framework that represents intermediate visual thoughts in a fixed-size discrete cosine space. By exploiting strong energy compaction, *SpecFlow* preserves global layout and relational structure while introducing high-frequency details only when increased spatial precision is required. To align visual state evolution with linguistic intent, classifier-free guidance enables autoregressive textual thoughts to steer flow-based updates of the visual workspace (state) without expanding the context. As a result, *SpecFlow* maintains a bounded visual workspace whose updates depend only on the current visual state and accumulated textual trace, enabling long-horizon inference with stable latency and memory usage independent of reasoning depth. Empirical results show that *SpecFlow* achieves competitive or superior reasoning performance while reducing computation and KV cache costs by up to $2.1\times$.

## 1. Introduction

Autoregressive multimodal spatial reasoning (Li et al., 2025a; Wu et al., 2024b) extends the chain-of-thought paradigm (Wei et al., 2022) by interleaving textual and vi-

[1]Informatics Institute, University of Amsterdam, Amsterdam, The Netherlands [2]University of Thessaly, Volos, Greece [3]Department of Computer Science, University College London [4]Amazon AGI, Seattle, USA [5]Department of Electronic and Electrical Engineering, Trinity College Dublin, Dublin, Ireland [6]School of Information Technology, Halmstad University, Halmstad, Sweden. Correspondence to: Anuj Pathania <a.pathania@uva.nl>.

*Proceedings of the 43$^{rd}$ International Conference on Machine Learning*, Seoul, South Korea. PMLR 306, 2026. Copyright 2026 by the author(s).

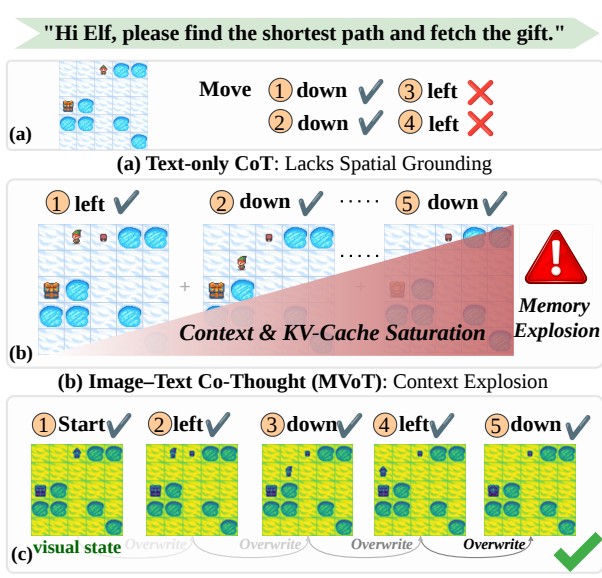

*Figure 1.* Comparison of multimodal spatial reasoning paradigms. (a) Text-only CoT lacks spatial grounding; (b) Image–Text Co-Thought improves grounding but causes context and KV-cache growth; (c) *SpecFlow* updates a compact spectral visual state, enabling efficient multi-hop reasoning with stable memory.

sual tokens, which allows the visual tokens ground linguistic reasoning in spatial structure and provides essential cues for tracking layouts, object relationships, and state transitions over time. This interleaved generation paradigm enables the joint reasoning through language and explicit visual representation, resolving ambiguities that arise in purely textual reasoning. It has demonstrated prevailing effectiveness to support more robust reasoning, memory, and decision-making, especially in spatial and visuospatial tasks (Hu et al., 2024).

However, this paradigm usually requires multi-hop inference, in which the intermediate thoughts (e.g., object layouts, and spatial configurations) are explicitly constructed and reused across successive reasoning steps. The repeated generation, which is appended to the context, externalizes the evolving spatial state and leads to the monotonic growing of the sequence length, particularly dominant in the visual modality. For example, a single MVoT-style intermediate thought can consist of $\mathcal{O}(10^3)$ visual tokens, much more than the corresponding textual tokens (i.e., typically fewer

than $\mathcal{O}(10^2)$). This can cause the attention computation and key-value caching cost to increase rapidly as the number of reasoning hops grows. Unfortunately, such growth, especially in long-horizon inference, quickly exhausts the available context window, amplifies memory bandwidth pressure, and leads to high latency (Li et al., 2025a).

Prior work that alleviates context growth can be broadly grouped into two lines. (1) *Explicit token pruning*. Visual token compression methods (Zhang et al., 2025d; Chen et al., 2024) prune image tokens at inference time, but typically rely on local, heuristic importance criteria and do not account for the dynamics of multi-hop reasoning, where spatial cues must be reused across steps. As a result, single-thought-based pruning may either discard necessary global structure or retain redundant tokens, leading to under- or over-pruning. (2) *Implicit latent reasoning* methods (Shen et al., 2025a; Su et al., 2025) internalize intermediate thoughts into continuous latent states, but these representations often lack explicit supervision and controllability and typically underperform approaches that externalize visual thoughts for complex spatial reasoning.

We address this bottleneck by rethinking the role of intermediate visual thoughts. In contrast to dense visual inputs, intermediate thoughts in spatial reasoning often only need to carry sparse, abstract cues such as global layout and geometric relations, rather than pixel-level detail (Xu et al., 2025b; Cheng et al., 2024). Directly generating pixel-space thoughts therefore introduces unnecessary degrees of freedom and cost. Our goal is to represent intermediate thoughts as a compact evolving spatial state that can be consumed by subsequent hops. To this end, we generate intermediate visual thoughts using diffusion-based generation. Efficient multi-hop reasoning, however, requires stable and deterministic updates with few solver steps. We instantiate flow matching to learn a velocity field that generates each intermediate visual thought. Nevertheless, each thought still requires multi-step sampling, which incurs nontrivial computation and slows down multi-hop inference.

To further enable lightweight multi-hop reasoning, we design flow matching directly in the discrete cosine projection space. Cosine projection is an orthogonal transformation that concentrates most signal energy into low-frequency coefficients. As shown in Figure 3, after projecting visual inputs into the cosine domain (Du et al., 2025), low frequencies primarily encode global layout, while high frequencies capture fine details that are often incidental to reasoning. This separation allows a compact spectral budget that prioritizes low-frequency structure and activates higher frequencies only when additional spatial detail is required. Flow matching then transports the frequency-limited visual state in cosine space, updating only the active coefficients conditioned on the current textual thought and the previous visual

state. With classifier-free guidance, the autoregressive textual trace serves as a control signal that steers the flow-based update of the visual workspace, aligning linguistic intent with spatial evolution without appending visual tokens to the context. Concretely, we propose Spectral Progressive Thought Flow (*SpecFlow*), which maintains a fixed-size visual workspace whose update depends only on the current state, allowing earlier visual thoughts to be discarded. As a result, memory and computation scale with the textual trace plus a constant-size visual workspace, enabling long-horizon multimodal spatial reasoning with stable memory.

We summarize our main contributions as follows:

- We introduce a new multimodal spatial reasoning paradigm that enables parallel visual workspace (state) evolution, eliminating sequential visual token generation and the resulting token accumulation, and decoupling memory usage from reasoning depth.

- We formulate intermediate visual thought evolution as a deterministic flow-matching process in cosine space, enabling stable and lightweight hop-wise state updates with only a small and fixed number of solver steps while preserving coarse-to-fine spatial structure.

- We integrate classifier-free guidance to steer the flow-based visual updates, allowing long-horizon inference with *stable memory usage* that scales primarily with the textual trace rather than the number of visual hops.

## 2. Related work

**Multimodal Spatial Reasoning (MSR).** Existing spatial reasoning approaches can be broadly grouped into three paradigms. First, *two-stage abstraction methods* convert visual observations into structured symbolic representations, such as text (Zhang et al., 2024), graphs (Mitra et al., 2024; Mondal et al., 2024), or bounding boxes (Lei et al., 2024) before performing reasoning. Second, *tool-augmented methods* integrate external components, such as planners, solvers, or verifiers, to support multi-hop reasoning over complex visual scenes (Yao et al., 2023; Yang et al., 2023; Hu et al., 2024; Zhou et al., 2024; Li et al., 2024; Gao et al., 2024). Third, *unified sequence methods* (Li et al., 2025b;a) directly interleave visual and textual tokens within a single autoregressive stream. While unified models are conceptually appealing, they typically externalize intermediate visual thoughts as long token sequences, which rapidly increases context length and KV-cache footprint, thereby limiting reasoning depth and inference efficiency.

**Visual Token Compression.** Existing approaches can be categorized into three paradigms. First, *training-based methods* (Zhang et al., 2025c; Tong et al., 2024; Raposo et al., 2024) incorporate token reduction directly into model ar-

chitectures or training objectives, but often overlook the structured flow of visual information across reasoning hops. Second, *training-free methods* (Chen et al., 2024; Zhang et al., 2025d; Tan et al., 2025) perform token pruning at inference time, but typically relying on heuristic importance criteria that are not optimized for multi-hop reasoning. Third, *continuous latent reasoning* approaches (Hao et al., 2024; Cheng & Van Durme, 2024; Deng et al., 2024; Xu et al., 2025a; Shen et al., 2025a; Su et al., 2025) project intermediate reasoning states into compact continuous latent spaces. However, these latent representations usually lack explicit supervision and interpretability, making it difficult to align latent transitions with discrete spatial states, which limits their applicability to multi-hop multimodal spatial reasoning.

**Flow Matching and Efficient Generative Transport.** Flow based generative modeling learns the deterministic transport dynamics from a base distribution to a target distribution. It enables stable sampling with a small number of integration steps and substantially reduces the iterative overhead of stochastic diffusion (Albergo & Vanden-Eijnden, 2022; Liu et al., 2022; Lipman et al., 2022). In parallel, latent diffusion models (Rombach et al., 2022) have become a standard strategy for scalable high fidelity generation. More recently, multimodal diffusion transformers (Esser et al., 2024) have further advanced generative modeling by unifying transformer based architectures with diffusion style objectives. Despite these advances, prior work primarily targets one-shot generation or generic synthesis, and does not directly address the multi-hop setting where a model must repeatedly generate intermediate visual thoughts under tight memory and latency constraints.

# 3. Proposed Method

## 3.1. Problem Formulation

Multimodal spatial reasoning (MSR) updates a coherent spatial state and leverages linguistic evidence to perform multi-hop decision-making. Let $x^{\text{vis}}$ and $x^{\text{txt}}$ denote the input of an initial visual observation (e.g., image, and map) and an initial textual instruction or query. MSR can require multi-hop inference. A *hop* is defined as a reasoning cycle that updates a visual thought capturing the current spatial state and then produces the corresponding textual thought.

Different from MVoT-style approaches that autoregressively generate and append dense visual tokens to the context, as depicted in Figure 2, we formulate the visual modality as a *fixed-size continuous state* that is *overwritten* at each hop and used only for conditioning. Let $\hat{t}_i$ and $\hat{v}_i$ denote the generated textual and visual thoughts at hop $i$, respectively. We design a hybrid process in which textual thoughts are generated autoregressively. The visual thought is produced

via a *state transition* conditioned on the current textual context and the most recent visual state:

$$\hat{v}_{i+1} \sim \mathcal{P}_\theta\big(v_{i+1} \mid x^{\text{vis}}, x^{\text{txt}}, \hat{t}_{\leq i}, \hat{v}_i\big), \quad (1\text{a})$$

$$\hat{t}_{i+1} \sim \mathcal{P}_\theta\big(t_{i+1} \mid x^{\text{vis}}, x^{\text{txt}}, \hat{t}_{\leq i}, \hat{v}_{i+1}\big), \quad (1\text{b})$$

where $\hat{t}_{\leq i} = \{\hat{t}_1, \ldots, \hat{t}_i\}$. Eq. (1a) imposes a Markov property on the visual stream. The next visual thought depends on the latest visual state and the accumulated textual trace, but does not require all past visual thoughts in the context. Eq. (1b) keeps language reasoning autoregressive and conditions each new textual thought on the updated visual state, so that spatial cues directly influence subsequent reasoning.

## 3.2. Spectral-Progressive Frequency Allocation

Multimodal spatial reasoning can involve multi-hop inference. The generation of intermediate visual thoughts across hops can lead to the monotonic growing of sequence length and computational cost, particularly for the visual modality with dense pixels. Block cosine transformation exhibits a strong energy compaction, where a large fraction of signal energy concentrates in a small set of low-frequency coefficients. Retaining only the low-frequency components is often sufficient to recover the global layout and coarse spatial structure of the scene, as illustrated in Fig. 3. In contrast, the high-frequency components primarily encode appearance variations, which are less critical for multi-hop spatial reasoning and can therefore be deferred to later stages.

Concretely, we employ a block cosine projection to the visual state. Let $\mathcal{D}_b(\cdot)$ denote the forward projection with a block size $b \times b$ and let $\mathcal{D}_b^{-1}(\cdot)$ denote its inverse. We allocate a progressively expanding spectral budget over the flow time $t \in \{0, 1, \ldots, T\}$ using a binary mask $M_t \in \{0, 1\}^{b \times b}$ over intra-block frequencies, which is shared across blocks. The number of active frequency modes is

$$m(t) = \sum_{u=0}^{b-1} \sum_{v=0}^{b-1} M_t(u, v), \quad (2)$$

where $m(t)$ is chosen to be non-decreasing in $t$. At early times, $M_t$ retains only a small low-frequency region (e.g., the DC term and a few neighboring modes), yielding a compact representation that preserves coarse geometry while discarding fine detail. As $t$ increases, $M_t$ progressively unblocks mid- and high-frequency bands to enable refinement. Given an image $x$, the frequency-limited reconstruction at time $t$ is

$$\hat{x}_t = \mathcal{D}_b^{-1}\Big(M_t \odot \mathcal{D}_b(x)\Big), \quad (3)$$

where $\odot$ denotes element-wise multiplication with broadcasting of $M_t$ over blocks. Eq. (3) indicates that early states are structure-dominant, while later states become progressively sharper as more frequencies are activated.

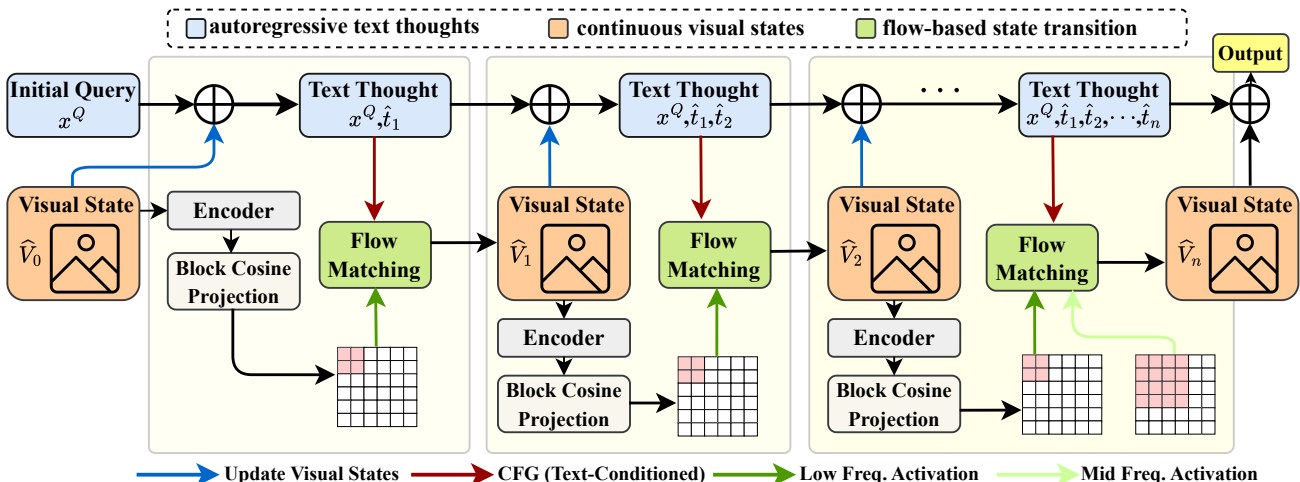

*Figure 2.* Spectral-Progressive Thought Flow (*SpecFlow*) alternates autoregressive text thoughts with text-conditioned, flow-based updates of a continuous visual state. Visual states are overwritten at each hop and represented in the cosine domain with progressively activated frequency bands, enabling efficient multimodal spatial reasoning without accumulating visual tokens or growing the context length.

This spectral schedule induces a hierarchical coarse-to-fine curriculum aligned with multi-hop reasoning. Specifically, low frequencies stabilize spatial relations, and high frequencies add texture only after the configuration is consistent. In addition to a fixed schedule, the spectral budget can be made text-adaptive by mapping the current textual context to a target budget. We keep this mechanism lightweight and defer its details to the text-guidance formulation in Sec. 3.4.

### 3.3. Cosine-Space Flow Matching

The visual thought is invoked repeatedly across hops, so its per-hop cost must be tightly controlled. A stable deterministic inference within a small number of solver steps is therefore demanded. *SpecFlow* resolves this by performing flow matching directly in the discrete cosine space under a spectral mask, yielding smoother dynamics and requiring fewer solver steps. Specifically, the early integration steps operate on a small subset of low-frequency coefficients that capture global layout, while higher-frequency coefficients are activated only when fine details are necessary.

Building on the cosine-domain visual representation introduced in Sec. 3.2, we model the evolution of visual thoughts as a deterministic continuous-time flow in coefficient space. For a continuous time variable $t \in [0, 1]$, we define a coefficient trajectory $X(t)$, governed by

$$\frac{dX(t)}{dt} = u_\theta\left(\widetilde{X}(t), t, c\right), \quad (4)$$

where $u_\theta$ is a learnable velocity field and $c$ denotes the conditioning signal (e.g., the current textual thought). The state fed to the velocity field is spectrally filtered as

$$\widetilde{X}(t) = M(t) \odot X(t), \quad (5)$$

where $M(t) \in \{0, 1\}^{b \times b}$ is a mask schedule that expands the active intra-block frequency set from low to high frequencies as $t$ increases, and $\odot$ denotes element-wise multiplication with broadcasting over blocks. This construction ensures that early dynamics focus on coarse structure and later dynamics refine higher-frequency details.

**Flow matching objective in cosine space.** To train $u_\theta$, we adopt a standard flow-matching objective defined by end-point pairs. Let $x_0$ be a data sample and let $x_1 \sim \mathcal{N}(0, I)$ be a noise sample of the same shape. We transform both endpoints into cosine coefficients, $X_0 = \mathcal{D}_b(x_0)$ and $X_1 = \mathcal{D}_b(x_1)$, and sample an intermediate point by linear interpolation in coefficient space:

$$X_t = (1 - t) X_1 + t X_0, \quad (6)$$

which induces the constant target velocity via

$$\dot{X}_t \triangleq \frac{dX_t}{dt} = X_0 - X_1. \quad (7)$$

We train the model to predict this velocity from the masked intermediate state and time index:

$$\mathcal{L}_{\text{FM}} = \mathbb{E}_{x_0, x_1, t}\left[\left\|u_\theta(M(t) \odot X_t, t, c) - (X_0 - X_1)\right\|_2^2\right]. \quad (8)$$

Because the coefficients are explicitly ordered by frequency, $M(t)$ exposes only the currently active bands and the learned dynamics are naturally coarse-to-fine. When $M(t)$ retains only low frequencies, the model learns to rapidly establish global layout. As $M(t)$ expands, the model learns to refine mid- and high-frequency structure.

**Fast deterministic inference with spectral constraints.** At inference time, each hop is initialized from the current

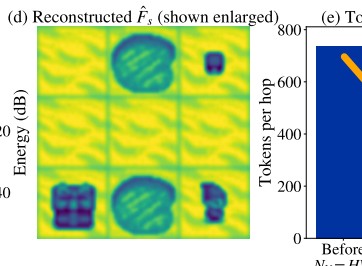

(a) Original visual state $F_s$  (b) Partition into $b \times b$ blocks  (c) Avg. cosine energy (freq domain)  (d) Reconstructed $\hat{F}_s$ (shown enlarged)  (e) Token count reduction

*Figure 3.* **Spectral-progressive frequency allocation with block cosine projection.** (a) An intermediate visual state. (b) Partition into $b \times b$ blocks and employ block cosine projection. (c) Average coefficient energy concentrates in low frequencies. (d) Reconstruction using only the low-frequency bands preserves global layout. (e) Retaining a small subset of coefficients yields significant visual-token reduction per hop.

visual workspace rather than from fresh noise. Specifically, at hop $i$, we project the previous visual state into the cosine domain as $X_i^{(0)} = \mathcal{D}_b(\hat{v}_i)$, and integrate the text-guided ODE in Eq. (4) from $t = 0$ to $t = 1$ using a fixed-step solver, such as Euler. The resulting coefficient state $X_i^{(1)}$ is then mapped back by the inverse cosine transform to obtain the updated visual workspace, computed as $\hat{v}_{i+1} = \mathcal{D}_b^{-1}(X_i^{(1)})$.

As only a subset of frequency bands is active at any given time, the high-quality samples can therefore be obtained with a small number of solver steps. Consequently, each hop updates the overwritten visual workspace efficiently while respecting the evolving spectral budget defined by $M(t)$. The stable and coherent thought flow sharpens progressively over the integration time, fitting multi-hop reasoning.

### 3.4. Text-Guided Conditioning via CFG

Cosine-space flow matching specifies *how* the visual workspace evolves under spectral constraints, but multi-hop reasoning also requires controlling *what* the visual thoughts depict to align with the current textual intent. We therefore condition the velocity field on a hop-specific text context. Let $c_i$ denote the conditioning signal at hop $i$, which expresses the original query together with the generated textual trace up to hop $i$. To enable classifier-free guidance (CFG), we train the same velocity model to handle both conditional and unconditional inputs by randomly dropping the condition during training. With probability $p_{\text{drop}}$, we replace $c_i$ with a null token $\varnothing$, so the model learns both $u_\theta(\cdot, t, c_i)$ and $u_\theta(\cdot, t, \varnothing)$.

At inference time, we strengthen the influence of text by combining conditional and unconditional velocities at the *velocity level*. Given the masked coefficient state $\widetilde{X}(t) = M(t) \odot X(t)$, we form the guided velocity as

$$
\begin{aligned}
u_\theta^{\text{guid}}(X(t), t; c_i) &= u_\theta(M(t) \odot X(t), t, \varnothing) \\
&+ w\, u_\theta(M(t) \odot X(t), t, c_i) \\
&- w\, u_\theta(M(t) \odot X(t), t, \varnothing).
\end{aligned} \tag{9}
$$

where $w > 0$ is the guidance scale. The difference term isolates the text-attributable direction of change in the velocity field, and scaling it increases semantic compliance while retaining the unconditional prior for stability. We then integrate the guided ODE via

$$
\frac{dX(t)}{dt} = u_\theta^{\text{guid}}(X(t), t; c_i) \tag{10}
$$

from $t = 0$ to $t = 1$ to obtain the updated coefficient state $X(1)$ for hop $i$. The corresponding pixel-space visual thought is reconstructed by the inverse discrete cosine transform, given by $v_i = \mathcal{D}_b^{-1}(X(1))$.

Finally, the guided visual thought $v_i$ provides explicit spatial cues for the next language update. After generating $v_i$ at hop $i$, we generate the next textual thought autoregressively conditioned on the textual history and the current visual state, given by

$$
t_{i+1} \sim \mathcal{G}\big(t \,\big|\, x^{\text{txt}}, t_{\leq i}, v_i\big). \tag{11}
$$

In this way, text steers the cosine-space flow via CFG to produce a spatially grounded visual thought, and the resulting visual state in turn conditions subsequent text, improving cross-hop consistency.

## 4. Experiments

**Benchmark.** We evaluate *SpecFlow* on a diverse set of benchmarks covering multi-hop multimodal spatial reasoning and spatial decision-making (see Appendix E). These benchmarks span varying reasoning depths and spatial complexities, enabling a systematic evaluation of long-horizon reasoning accuracy and robustness under increasing difficulty. For more details of these benchmarks, please refer to Appendix A. For all benchmarks, we adopt the official task definitions and difficulty splits established in prior work (Li et al., 2025a;b).

**Model Specification.** *SpecFlow* couples a flow-matching (FM) *visual-thought* generator in VAE latent space with an

*Table 1.* Comparison on multimodal spatial reasoning. $\uparrow$ indicates higher is better; $\downarrow$ indicates lower is better. *SpecFlow* achieves competitive accuracy while significantly reducing FLOPs, latency, and memory

| Methods | Compositional tasks, e.g., spatial reasoning and visual search | | | | | | | |
|---|---|---|---|---|---|---|---|---|
| | VSR (Liu et al., 2023) | | | | V-Star (Wu & Xie, 2024) | | | |
| | Acc.(%)$^\uparrow$ | FLOPs(G)$^\downarrow$ | Lat.(s)$^\downarrow$ | Mem.(GB)$^\downarrow$ | Acc.(%)$^\uparrow$ | FLOPs(G)$^\downarrow$ | Lat.(s)$^\downarrow$ | Mem.(GB)$^\downarrow$ |
| VoCoT | 68.88 | 20342.40 | 0.65 | 56.37 | 59.87 | 22334.01 | 0.71 | 59.28 |
| *SoT* | 54.56 | 17675.04 | 0.59 | 52.62 | 38.06 | 20319.23 | 0.66 | 56.53 |
| *LightFastV* | 58.63 | 16280.49 | 0.55 | 49.65 | 46.83 | 18802.51 | 0.60 | 53.32 |
| *SparseVLM* | 63.88 | 15200.29 | 0.50 | 45.75 | 57.36 | 17802.00 | 0.54 | 51.66 |
| *Heima* | 51.69 | **10394.46** | **0.40** | **38.84** | 41.70 | 14314.26 | **0.42** | **40.62** |
| *PCCoT* | 54.71 | 10654.00 | 0.42 | 39.79 | 44.40 | **13963.49** | 0.50 | 42.31 |
| *CODI* | 68.82 | 11099.76 | 0.46 | 39.02 | 48.43 | 14279.39 | 0.48 | 42.65 |
| ***SpecFlow*** | **70.14** | 11169.74 | 0.41 | 39.53 | **61.28** | 13985.48 | 0.45 | 41.22 |

| Methods | EmbSpatial (Du et al., 2024) | | | | Winoground (Thrush et al., 2022) | | | |
|---|---|---|---|---|---|---|---|---|
| | Acc.(%)$^\uparrow$ | FLOPs(G)$^\downarrow$ | Lat.(s)$^\downarrow$ | Mem.(GB)$^\downarrow$ | Acc.(%)$^\uparrow$ | FLOPs(G)$^\downarrow$ | Lat.(s)$^\downarrow$ | Mem.(GB)$^\downarrow$ |
| VoCoT | 59.76 | 27211.34 | 0.74 | 61.52 | 70.09 | 28092.92 | 0.80 | 62.23 |
| *SoT* | 56.14 | 24784.89 | 0.68 | 58.05 | 64.48 | 26639.10 | 0.74 | 60.16 |
| *LightFastV* | 58.32 | 23482.94 | 0.64 | 55.11 | 65.83 | 25253.62 | 0.69 | 56.75 |
| *SparseVLM* | 63.89 | 20785.15 | 0.74 | 51.93 | 66.01 | 23738.55 | 0.79 | 52.88 |
| *Heima* | 54.66 | **17607.64** | **0.42** | 43.78 | 65.64 | 18259.34 | 0.50 | **43.09** |
| *PCCoT* | 56.13 | 17719.32 | 0.49 | 43.84 | 67.49 | **17556.18** | **0.49** | 53.33 |
| *CODI* | 55.74 | 18584.17 | 0.44 | 44.22 | 68.91 | 17985.55 | 0.50 | 50.21 |
| ***SpecFlow*** | **67.79** | 17731.50 | 0.44 | **42.57** | **70.47** | 18390.14 | **0.49** | 46.94 |

*Table 2.* Performance comparison on spatial decision-making tasks across increasing environment sizes. All values denote success rate (%). GRPO results are reported for open-source MLLMs following prior work.

| Model | Setting | Maze (Ivanitskiy et al., 2023) | | | | MiniBehavior (Jin et al., 2023) | | | | FrozenLake (Wu et al., 2024a) | | | | Avg |
|---|---|---|---|---|---|---|---|---|---|---|---|---|---|---|
| | | 4 | 8 | 12 | 16 | 8 | 12 | 16 | 20 | 4 | 8 | 12 | 16 | |
| *Closed-Source MLLMs* | | | | | | | | | | | | | | |
| Gemini-3-Flash | Naive | **94.38** | 91.12 | 76.94 | 61.27 | **92.71** | 90.36 | 78.02 | 70.19 | 86.58 | **81.74** | 62.31 | 46.42 | 77.67 |
| GPT-5.1 | Naive | 91.76 | 88.94 | 72.83 | 58.91 | 89.82 | 87.45 | 73.96 | 66.14 | 83.27 | 79.68 | 58.74 | 42.85 | 74.53 |
| *Open-Source MLLMs* | | | | | | | | | | | | | | |
| Qwen3-VL-8B | Naive | 61.72 | 42.75 | 31.39 | 21.07 | 63.74 | 46.28 | 34.41 | 23.96 | 55.21 | 37.25 | 22.37 | 17.21 | 38.11 |
| Qwen3-VL-8B | SFT | 78.92 | 61.47 | 48.63 | 36.85 | 80.34 | 63.18 | 50.27 | 38.94 | 71.46 | 53.72 | 37.86 | 29.41 | 54.25 |
| Qwen3-VL-8B | GRPO | 75.31 | 57.88 | 44.92 | 33.27 | 76.85 | 59.41 | 46.38 | 35.02 | 68.02 | 50.21 | 34.56 | 26.98 | 50.73 |
| Qwen3-VL-32B | Naive | 72.18 | 54.36 | 41.92 | 29.84 | 74.05 | 56.71 | 44.28 | 31.63 | 65.47 | 48.12 | 33.06 | 24.58 | 48.02 |
| Qwen3-VL-32B | SFT | 86.21 | 70.84 | 58.97 | 46.35 | 87.43 | 72.16 | 60.41 | 48.72 | 79.58 | 62.91 | 47.36 | 38.24 | 63.27 |
| Qwen3-VL-32B | GRPO | 83.02 | 67.31 | 55.26 | 42.17 | 84.18 | 69.04 | 56.78 | 44.35 | 76.43 | 59.47 | 44.02 | 35.91 | 59.83 |
| *Multimodal Interleave Reasoning* | | | | | | | | | | | | | | |
| MVoT | Autoregressive | 92.03 | 88.61 | 72.94 | 57.81 | 89.14 | 86.78 | 73.88 | 65.23 | 83.31 | 78.94 | 58.62 | 41.87 | 74.10 |
| DiffThinker | Flow(image) Only | 93.10 | 89.95 | 74.86 | 59.42 | 90.82 | 88.31 | 75.21 | 66.58 | 84.96 | 79.83 | 60.41 | 43.62 | 75.59 |
| *SpecFlow* (Ours) | Flow(image) + AR(Text) | 94.12 | **92.04** | **78.31** | **64.85** | 92.66 | **91.27** | **79.84** | **72.96** | **87.94** | 81.73 | **64.72** | **49.38** | **79.15** |

autoregressive (AR) text module. The VAE latent is transformed to cosine space and progressively activated by a time-dependent spectral mask $M(t)$ from low to high frequencies. At each reasoning hop, the AR module produces a compact textual thought/plan to condition FM generation, while the visual thought is generated non-autoregressively via flow matching. Unless stated otherwise, SpecFlow uses Qwen3-VL-8B together with QwenImage-Edit-2509 (20B) (Esser

et al., 2024; Wu et al., 2025a). To exclude confounding effects from parameter scaling, we additionally report strong AR baselines with Qwen3-VL-32B.

**Baselines.** We compare *SpecFlow* against baselines spanning *prompt-based*, *heuristic*, and *latent-space* token-compression paradigms, including *SoT* (Aytes et al., 2025); *LightFast*, which combines *LightThinker* (Zhang et al., 2025b) for textual pruning and *FastV* (Chen et al., 2024)

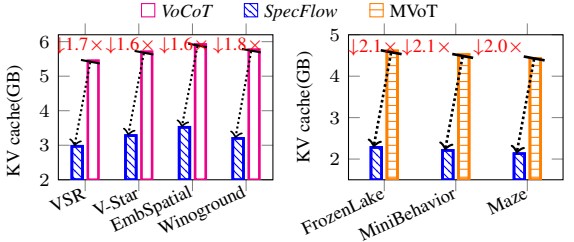

(a) Multimodal spatial reasoning   (b) Dynamic multimodal spatial reasoning

*Figure 4.* KV-cache memory across benchmarks. *SpecFlow* achieves 1.6×–2.1× KV-cache reduction over baselines.

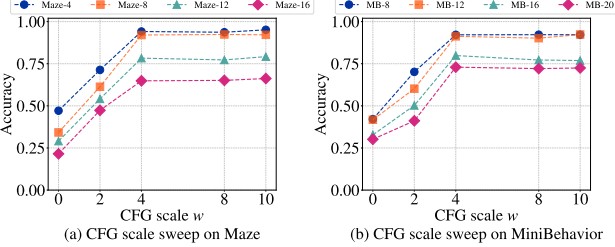

(a) CFG scale sweep on Maze   (b) CFG scale sweep on MiniBehavior

*Figure 5.* Effect of CFG guidance scale on reasoning accuracy, Performance improves with increasing guidance strength and peaks at a moderate scale ($w = 4$), while excessively large guidance yields diminishing returns due to over-deterministic conditioning.

for visual pruning; and latent reasoning approaches such as *Heima* (Shen et al., 2025a), *CODI* (Shen et al., 2025b), and *PCCoT* (Wu et al., 2025b). We further include *SparseVLM* (Zhang et al., 2025d), which iteratively sparsifies visual tokens to reduce computation, as well as *DiffThinker* (He et al., 2025), which performs reasoning purely in the image domain.

**Training and Inference Settings.** We use a flow-matching objective with linear interpolation between data latents and Gaussian noise, MSE velocity supervision and classifier-free guidance (CFG) applied to the predicted velocity. We adopt a first-order Euler ODE solver with a fixed number of steps $T=5$ and CFG scale $w=4$, and sample time steps from a logit-normal distribution. The FM model is fine-tuned using LoRA (rank 32) for 5 epochs with learning rate $1\times10^{-4}$. For AR supervised fine-tuning (SFT), we use 5 epochs with learning rate $1\times10^{-4}$, LoRA rank 32; for GRPO, we train for 1 epoch with learning rate $1\times10^{-6}$, rollout size $n=4$, KL coefficient $1\times10^{-2}$.

### 4.1. Results on Compositional Multimodal Reasoning

Table 1 summarizes results on four spatially grounded benchmarks covering claim verification (**VSR** (Liu et al., 2023)), target-centric search (**V-Star** (Wu & Xie, 2024)), relational grounding in complex scenes (**EmbSpatial** (Du et al., 2024)), and compositional image–caption alignment (**Winoground** (Thrush et al., 2022)). Across these settings, *SpecFlow* matches or exceeds the best-performing baselines while operating in a substantially lighter inference regime. On VSR and V-Star, *SpecFlow* attains the top ac-

curacy and reduces both latency and memory by roughly one-third relative to VoCoT, avoiding the long-context cost of prompt-heavy interleaving. Compared with lightweight latent-reasoning approaches such as Heima and PCCoT, *SpecFlow* operates in a similar low-latency regime but yields markedly higher accuracy, with +18.5 on VSR over Heima and +16.9 on V-Star over PCCoT. On more challenging embodied and grounding benchmarks, *SpecFlow* improves over the strongest prior baseline on EmbSpatial by a clear margin (67.79 vs. 63.89 of SparseVLM), and also surpasses VoCoT on Winoground (70.47 against 70.09), and, crucially, these gains come with low latency and reduced memory footprint with faster execution time.

### 4.2. Results on Spatial Decision Making

Table 2 compares *SpecFlow* with both open- and closed-source baselines on three spatial decision-making benchmarks. On **Maze** (Ivanitskiy et al., 2023), *SpecFlow* ranks first across all grid sizes; the advantage becomes most visible at larger layouts, reaching 64.85% on size 16, where autoregressive baselines degrade sharply with horizon length. **MiniBehavior** (Jin et al., 2023) shows a similar scaling pattern: *SpecFlow* remains strong at the largest environment, achieving 72.96% on size 20, which maintains object-centric subgoals over long action sequences. On **FrozenLake** (Wu et al., 2024a), where early mistakes cascade under sparse-reward navigation, *SpecFlow* again leads and improves over MVoT as well as Qwen3-VL baselines trained with SFT or GRPO. Compared with image-only flow reasoning, the gains are clearer at larger layouts: against DiffThinker, *SpecFlow* improves the average success from 75.59 to 79.15, and raises Maze-16 from 59.42 to 64.85. Notably, these gains are not explained by AR scale: with a smaller AR backbone than Qwen3-VL-32B, *SpecFlow* attains a higher average success rate, 79.15 vs. 63.27, under the same evaluation protocol.

### 4.3. KV-cache Efficiency

Figure 4 reports KV-cache memory consumption. On compositional multimodal spatial reasoning tasks, *SpecFlow* consistently reduces KV-cache usage to approximately 3.0–3.5 GB, compared to 5.4–5.9 GB for VoCoT, corresponding to a 1.6×–1.8× reduction. This improvement stems from its non-autoregressive visual-thought generation, which prevents KV growth across reasoning steps. The advantage is larger on dynamic spatial decision-making tasks; *SpecFlow* reduces KV-cache by about 2.1× relative to MVoT and stays near 2.1–2.3 GB across tasks. Notably, these gains are consistent across benchmarks with varying reasoning depth and modality interaction patterns, indicating that its KV efficiency is not task-specific but intrinsic to its flow-based generation paradigm.

*Table 3.* Effect of discrete 3-band spectral scheduling. Accuracy ↑
, latency(min) ↓ and FLOPS(G)↓ are reported per episode.

| Environment | Schedule | Accuracy ↑ | Latency ↓ | FLOPS(G) ↓ |
|---|---|---|---|---|
| | Linear | **94.37** | 1.96 | 16752.23 |
| Maze | Cosine | 94.12 | 1.29 | 13976.71 |
| | Fixed | 90.39 | **1.13** | **12723.34** |
| | Linear | 87.79 | 2.97 | 18265.11 |
| FrozenLake | Cosine | **87.94** | 1.73 | 15991.07 |
| | Fixed | 82.37 | **1.42** | **13672.52** |

### 4.4. Ablation Studies

**Impact of Discrete Spectral Scheduling.** We ablate the
proposed spectral-progressive strategy by comparing linear,
cosine, and fixed schedules, where the fixed variant activates
only low-frequency components throughout generation. As
shown in Table 3, both progressive schedules outperform
the fixed baseline, demonstrating the necessity of gradu-
ally activating high-frequency components to refine visual
thoughts. Between progressive variants, the cosine sched-
uler achieves comparable or slightly better accuracy than
linear scheduling while substantially reducing inference la-
tency (up to 35% on Maze and 42% on FrozenLake), which
delays computationally expensive high-frequency activa-
tion. Although the fixed schedule yields the lowest latency,
its degraded accuracy highlights the balance between effi-
ciency and reasoning fidelity, validating our scheduling as a
favorable balance between performance and efficiency.

**Impact of Guidance Scale.** Figure 5 shows that reasoning
performance is robust across a wide range of classifier-free
guidance scales. Accuracy consistently improves as the
guidance scale increases from small values, peaking around
$w = 4$, where textual conditioning is sufficiently strong
to steer visual thought generation. When $w$ is too small,
conditioning is insufficient and leads to under-guided rea-
soning, while further increasing $w$ beyond this point yields
no additional gains and can even degrade performance due
to overly deterministic generation.

**Impact of Inference Steps.** Figure 6 analyzes the effect
of the number of ODE inference steps $T$ on reasoning per-
formance. Across both Maze and FrozenLake environments,
accuracy improves as $T$ increases from very small values,
indicating that insufficient integration steps lead to under-
refined visual thoughts. However, performance quickly satu-
rates around $T = 15$–$20$, beyond which additional steps pro-
vide only marginal gains while incurring noticeably higher
latency. These results suggest that a moderate number of
inference steps is sufficient to balance reasoning accuracy
and computational efficiency.

**Compound Effect of Spectral Projection & Latent VAE.**
Table 4 reveals a clear *compound* benefit when explicit spec-
tral scheduling is combined with a learned latent bottleneck

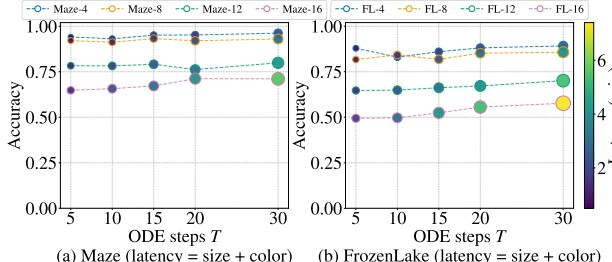

*Figure 6.* Effect of the number of ODE inference steps $T$ on rea-
soning accuracy. Marker size and color encode inference latency.
Accuracy improves with increasing $T$ and saturates at moderate
step counts, while larger $T$ leads to diminishing returns and higher
computational cost.

*Table 4.* **Ablation on spectral compression vs. learned bot-
tleneck.** We compare (i) *Spectral only*, which applies spectral
compression directly to the intermediate spatial layout without a
learned latent bottleneck; (ii) *VAE only*, which performs full VAE
encoding/decoding at every hop without spectral masking; and (iii)
*Spectral+VAE*, which performs spectral-progressive activation in
the VAE latent space. All settings use the same flow solver budget
of 5 inference steps per hop.

| Spectral | Latent VAE | FrozenLake | | EmbSpatial | |
|---|---|---|---|---|---|
| | | Acc.↑ | Lat.(min)↓ | Acc.↑ | Lat.(min)↓ |
| ✓ | ✗ | 81.47 | 1.97 | 64.37 | 3.43 |
| ✗ | ✓ | 87.15 | 6.33 | 66.02 | 10.31 |
| ✓ | ✓ | **87.94** | **1.29** | **67.79** | **2.35** |

under the same 5-step solver budget. Using *Spectral only*
is computationally efficient (1.97/3.43 min) but noticeably
less accurate (81.47/64.37), indicating that direct frequency
compression without a learned manifold can discard task-
critical semantics. Conversely, *VAE only* achieves strong
accuracy (87.15/66.02) yet substantially slower (6.33/10.31
min), reflecting the high per-hop overhead on the full latent
representation. Strikingly, *Spectral+VAE* attains the best
accuracy while reducing latency to 1.29/2.35 min, about
$4.9\times$ faster on FrozenLake and $4.4\times$ faster on EmbSpatial
than VAE-only, showing that spectral-progressive activation
*amplifies* the strengths of the VAE by making latent updates
compute-adaptive: early hops update low-frequency latent
structure where the flow is smoother and cheaper to inte-
grate, while later hops selectively unlock higher-frequency
components only when fine detail is needed.

## 5. Conclusion

We propose *SpecFlow*, a *new* lightweight framework that
replaces token-accumulating pixel-space visual thoughts
with a fixed-size, frequency-limited visual workspace in
the discrete cosine domain. By exploiting the energy com-
paction property of cosine representations, *SpecFlow* pre-
serves global layout and relational structure through low-
frequency components, while introducing higher-frequency
details only when increased spatial precision is required.
The visual workspace is updated via deterministic cosine-

space flow matching, which evolves only the active spectral coefficients conditioned on the current textual thought and visual state. With classifier-free guidance, visual updates remain aligned with textual reasoning, allowing earlier visual thoughts to be safely discarded without loss of essential spatial cues. As a result, *SpecFlow* decouples memory usage and computation from reasoning depth, enabling long-horizon multimodal spatial reasoning with stable latency, bounded memory, and effective spatial grounding, while reducing computation and KV cache costs by up to $2.1\times$.

## Acknowledgment

This work was funded under the COIN-3D project, which has received funding from the European Union's Horizon Europe research and innovation program under grant agreement No. 101159667.

## Impact Statement

This paper introduces a new mechanism for externalizing intermediate visual reasoning without saturating the multimodal context. Rather than denoising in pixel space or a learned latent space, SpecFlow evolves intermediate visual thoughts directly in the discrete cosine domain, maintaining a fixed-size spectral workspace throughout multihop inference. By coupling the generation process with frequency, early updates emphasize low-frequency coefficients to capture global layout and relational structure, while higher frequencies are activated only when additional spatial precision is required. In contrast to VAE-based latent diffusion, the cosine projection is a fixed and invertible transform without a learned bottleneck, enabling transparent, frequency-aware control over where computation is spent. As a result, SpecFlow reduces the cost of multi-hop multimodal spatial reasoning by preventing visual token accumulation and enabling deterministic updates with a small number of steps, improving practicality under tight memory and latency budgets. More broadly, this frequency-domain perspective highlights a controllable way to trade visual precision for compute while retaining interpretable spatial structure.

Looking forward, the same spectral, coarse-to-fine principle may extend beyond intermediate thoughts to efficient photorealistic diffusion generation. Progressively unlocking higher-frequency components provides a natural curriculum: establish global structure first, then synthesize fine-grained appearance. Such frequency-aware curricula could offer explicit control over realism–efficiency trade-offs and motivate future generative models that combine reasoning-efficient intermediate representations with high-fidelity visual synthesis.

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

# A. Dataset Details

## A.1. Spatial Reasoning Benchmarks.

To probe spatial grounding and compositional vision–language understanding, we consider four established benchmarks that emphasize complementary aspects of spatial reasoning (as shown in Table 5): (i) binary verification of visually grounded claims, (ii) fine-grained relational grounding in complex layouts, (iii) compositional image–caption alignment, and (iv) target-centric visual search in cluttered scenes. Unless otherwise noted, we follow each benchmark's official evaluation split and keep the original visual–textual inputs to avoid introducing confounding prompt effects.

**VSR** (Liu et al., 2023) provides image–text pairs where the text states a factual claim about the image; the task is to determine whether the claim is supported by visual evidence. We evaluate on the unseen test split. Following prior practice (Li et al., 2025b), we wrap each claim into a question-style prompt: *"Is there an event {description} in the image?"* to encourage explicit visual grounding.

**EmbSpatial** (Du et al., 2024) focuses on embodied spatial understanding with visually complex scenes and fine-grained spatial expressions (e.g., left/right relations between entities). We use the official test split and keep the original paired visual–textual inputs unchanged to faithfully measure grounding precision.

**V-Star** (Wu & Xie, 2024) evaluates visual search in cluttered environments, where the model must locate a target concept given a descriptive query. We follow the standard protocol and use the benchmark as-is, without introducing additional prompting.

**Winoground** (Thrush et al., 2022) is designed to test visio-linguistic compositionality through challenging image–caption matching: each sample contains two images and two subtly different captions. We formulate it as a caption selection problem, using the prompt *"Please describe the image."* to induce the model to select the caption that best matches each image.

*Table 5.* Spatial reasoning benchmarks used in our evaluation. We only specify additional prompting when it is explicitly applied; otherwise we keep the benchmark inputs unchanged.

| Benchmark | Primary capability | Eval split | Size | Input / prompting |
|---|---|---|---|---|
| VSR (Liu et al., 2023) | Claim verification | unseen test | 1222 | "Is there an event {description} in the image?" |
| EmbSpatial (Du et al., 2024) | Relational grounding | test | 3625 | No extra prompting; preserve original inputs |
| Winoground (Thrush et al., 2022) | Compositional alignment | test | 800 | Caption sel.: "Please describe the image. |
| V-Star (Wu & Xie, 2024) | Visual search | – | 238 | No extra prompting; use benchmark as-is |

## A.2. Dynamic Spatial Reasoning Tasks.

We evaluate *SpecFlow* on three dynamic spatial reasoning benchmarks, Maze (Ivanitskiy et al., 2023), MiniBehavior (Jin et al., 2023), and FrozenLake (Wu et al., 2024a), all of which require multi-step reasoning over simulated grid environments. These tasks stress (i) tracking time-varying spatial configurations, (ii) grounding action trajectories in the underlying layout, and (iii) inferring implicit goals under low-level visual dynamics.

Table 6 summarizes the core dataset characteristics used in our experiments, including grid size ranges, action-space properties, and dataset scale.

**Maze** (Ivanitskiy et al., 2023) is built with the Maze-Dataset framework, which generates 2D grid Mazes via an iterative depth-first search procedure. Mazes of sizes 3 to 16 are generated using multiple random seeds to diversify layout complexity. For each instance, a navigation path is constructed, and redundant or repeated paths are filtered out to reduce knowledge leakage between training and test splits. At evaluation time, the input consists of a Maze configuration together with three destination candidates (coordinate points), and the model must select the correct goal cell.

**MiniBehavior** (Jin et al., 2023) is derived from the INSTALLINGAPRINTER simulation suite. Reinforcement-learning agents are trained to complete procedural tasks in 7×7, 8×8, 12×12, 16×16, 20×20, grid environments using the Stable-Baselines3 library. The dataset includes diverse agent trajectories, and only successful action sequences (i.e., those that complete the simulated printer installation task) are retained. To discourage memorization, the dataset applies controlled environment perturbations: for repeated action paths or previously encountered environments, there is a 40% probability of perturbing either the printer or table coordinates and a 20% probability of removing one of these objects. After perturbation,

*Table 6.* Characteristics of dynamic spatial reasoning tasks, highlighting varying complexities in action dynamics and structural patterns.

| Characteristic | Maze | MiniBehavior | FrozenLake |
|---|---|---|---|
| Grid Sizes | 3,4,5,6, 8,12,16 | 7, 8,12,16,20 | 3,4,5,6,8,12,16 |
| Entity Types | 5 | 3 | 3 |
| Entities Numbers | 5 | 3 | 7.16 |
| Action Length | 14.33 | 17.92 | 15.91 |
| Action Types | 4 | 7 | 4 |
| Pattern Details | ✗ | ✗ | ✓ |
| Train Set Size | 10000 | 10000 | 10000 |
| Test Set Size | 2500 | 2500 | 2500 |

the original action sequence is replayed in the modified environment to confirm validity.

**FrozenLake** (Wu et al., 2024a) is adapted from OpenAI Gym's FrozenLake environment. It is a grid navigation setting in which an agent must reach a goal while avoiding holes, with action selection guided by Q-table-based policies. Trajectories are generated from agents acting greedily with respect to Q-values. Successful action sequences are included only if they have not appeared before in the same environment. For unsuccessful attempts (e.g., falling into a hole), trajectories are included with 50% probability either in their original form or with appended random actions. Ambiguous trajectories (neither clearly successful nor failed) are retained to increase coverage. Additionally, Q-tables are re-learned with randomly perturbed reward paths to introduce variance and mitigate overfitting.

We follow the same experimental conditions as MVoT (Li et al., 2025a). The three benchmarks above provide a complementary testbed for multi-step spatial-temporal reasoning, spanning procedurally generated layouts (Maze), procedurally structured tasks with controlled perturbations (MiniBehavior), and navigation under sparse-reward dynamics with pattern structure (FrozenLake, cf. Table 6).

# B. Implementation Details

This appendix details the experimental implementation of *SpecFlow* and the matched autoregressive (AR) baselines. We describe: (i) the flow-matching (FM) training objective and the deterministic ODE solver used for diffusion-based visual-thought generation, (ii) supervised fine-tuning (SFT) and GRPO settings for `Qwen3-VL` baselines, and (iii) the reward definitions and output-format constraints adopted to enable reliable automatic evaluation.

## B.1. Backbones and Training Protocol

***SpecFlow* backbones.** *SpecFlow* combines a diffusion-based visual-thought generator with an AR text pathway. For visual thoughts, we initialize from `Qwen-Image-Edit-2509` and fine-tune a **20B** Multimodal Diffusion Transformer (MMDiT) that operates in the latent space of a VAE. For textual reasoning and action emission, we use `Qwen3-VL-8B` as the default AR backbone. To control for model scaling in AR comparisons, we additionally report results with a stronger baseline, `Qwen3-VL-32B`, under the same training and evaluation protocol.

**Task-specific training and data fairness.** We train an FM model for each task category to avoid cross-task interference and to follow standard practice in task-specialized diffusion fine-tuning. All datasets are duplicated prior to training. For fairness, *SpecFlow* and AR baselines are trained on identical data distributions for each task, and all reported comparisons use the same train/validation/test splits.

**Autoregressive baselines.** AR baselines are instantiated with `Qwen3-VL-8B` and `Qwen3-VL-32B`. They are trained using SFT and, when applicable, further optimized with GRPO on the same task data used for the diffusion models. SFT prompts are designed to elicit concise answers (without explicit chain-of-thought outputs) to match our evaluation protocol. GRPO uses task-specific partial rewards to reduce the sparsity of binary exact-match supervision in long-horizon planning.

## B.2. Flow Matching for Generative Visual Thoughts

We employ flow matching to learn a continuous-time velocity field that deterministically transports a Gaussian noise latent to a solution-image latent under multimodal conditioning. The velocity field is parameterized by a Multimodal Diffusion Transformer and conditioned on both visual observations and textual instructions. At inference time, generation

*Table 7.* Hyperparameter settings for different training paradigms.

|  | **Flow Matching** | **SFT** | **GRPO** |
|---|---|---|---|
| Framework | DiffSynth-Studio (Zhang et al., 2025a) | SWIFT (Zhao et al., 2025) | verl (Sheng et al., 2025) |
| Epochs | 5 | 5 | 1 |
| Learning rate | $1 \times 10^{-4}$ | $1 \times 10^{-4}$ | $1 \times 10^{-6}$ |
| LoRA rank | 32 | 32 | – |
| Batch size | 4 | 16 | 64 (8B) / 32 (32B) |
| Rollout size ($n$) | – | – | 4 |
| KL coefficient | – | – | $1 \times 10^{-2}$ |

*Table 8. SpecFlow* instruction-tuning configuration.

| **Configuration** | **Instruction Tuning** |
|---|---|
| Visual Encoder | OpenAI-CLIP ViT-L/14 (Radford et al., 2021) |
| Backbone initialization | Qwen-Image-Edit-2509 (Esser et al., 2024; Wu et al., 2025a) |
|  | & Qwen3-VL-8B (Bai et al., 2023) |
| Optimizer | AdamW |
| Optimizer hyperparameters | $\beta_1 = 0.9,\ \beta_2 = 0.95,\ \epsilon = 10^{-4}$ |
| Global batch size | 64 |
| Peak learning rate (LLM) | $10^{-5}$ |
| Learning rate schedule | Cosine |
| Training epochs | 1 |
| Warm-up ratio | 0 |
| Weight decay | $3 \times 10^{-2}$ |
| Gradient clipping | 1.0 |
| Input image resolution | $336 \times 336$ |
| Max input length to LLM | 3072 |
| Numerical precision | bfloat16 |
| GPU usage | $4 \times$ (4 NVIDIA H100s) |
| Training time | 74.9h |

is performed by solving the resulting ordinary differential equation with a fixed-step solver, yielding deterministic visual-thought trajectories with predictable computational cost. This formulation avoids stochastic sampling and enables stable, low-variance generation suitable for multi-step reasoning.

### B.3. Training Hyperparameters

Table 7 summarizes the primary hyperparameters used for flow-matching fine-tuning, autoregressive supervised fine-tuning (SFT), and GRPO. For FM and SFT, we adopt identical learning rates and LoRA ranks to ensure comparable adaptation capacity across generative paradigms. GRPO is performed with a substantially smaller learning rate to stabilize policy optimization and prevent catastrophic drift from the supervised initialization. Batch sizes are selected to maximize hardware utilization under memory constraints, with larger effective batches used for GRPO to reduce variance across rollouts. The KL regularization coefficient follows standard practice in preference-based optimization to balance exploration and policy stability.

### B.4. Instruction Tuning Configuration

Table 8 reports the instruction-tuning configuration used to align the multimodal backbone with task-specific instructions. We initialize the visual and multimodal components from pretrained `Qwen-Image-Edit-2509` and `Qwen3-VL-8B` to preserve strong cross-modal representations while minimizing additional training cost. Instruction tuning is performed for a single epoch with a cosine learning-rate schedule and no warm-up, as the model is already well aligned and further training yields diminishing returns. We adopt conservative optimization settings (low peak learning rate, gradient clipping, and moderate weight decay) to stabilize fine-tuning and avoid overfitting. The input resolution and sequence length are chosen to match the maximum context required by downstream reasoning tasks, and all training is conducted in bfloat16 precision on NVIDIA H100 GPUs to balance numerical stability and efficiency.

### B.5. Prompting and Output Constraints for AR Baselines

To enable reliable automatic evaluation, all autoregressive baselines are constrained to emit a *single* final answer in a fixed, machine-parseable format, terminated by an explicit end-of-response marker. Intermediate reasoning traces, free-form explanations, or unconstrained continuations are disallowed. This design eliminates ambiguity in answer extraction, enforces consistent stopping behavior across models, and reduces evaluation variance arising from heterogeneous generation lengths. By standardizing output formats, we ensure that performance differences reflect reasoning capability rather than prompt sensitivity or decoding artifacts.

### B.6. Reward Functions for GRPO

Exact-match rewards are often too sparse for long-horizon planning, since a single early mistake can invalidate the entire trajectory. To provide denser learning signals, we define task-specific *partial* rewards that measure progress toward the correct solution.

**Sequential planning (Maze, MiniBehavior, FrozenLake).** Let $P = (p_1, \ldots, p_m)$ denote the predicted action sequence and $G = (g_1, \ldots, g_n)$ the ground-truth sequence. We reward the length of the longest correct prefix in $P$ relative to $G$:

$$R_{\text{plan}} = \frac{1}{n} \max \left\{ k \mid k \leq \min(m, n) \wedge \forall i \leq k, \ p_i = g_i \right\}. \tag{12}$$

This reward equals 1 only when the full ground-truth action sequence is matched, and decreases smoothly when errors occur earlier in the trajectory.

### B.7. Implementation of Baselines

We adopt three recent efficiency–oriented reasoning baselines, *SoT*, *LightFast*, and *Heima*, and align their settings with our *SpecFlow* evaluation pipeline.

**SoT (Sketch-of-Thought)** (Aytes et al., 2025) is a prompt-based approach that encourages models to express intermediate reasoning as compact "sketch" phrases, thereby reducing the number of generated tokens without any parameter updates. In our implementation, we inject the task-specific SoT directive ("`<sketch>`") before the original QWEN prompt. We keep the setting unchanged (temperature $0.7$). Since SoT operates purely through prompting, it introduces no extra training stage and does not alter the underlying model.

**LightFast (LightThinker + FastV)** (Zhang et al., 2025b; Chen et al., 2024) couples text-side and vision-side pruning in a modular manner: *LightThinker* reduces language-side overhead by removing low-utility reasoning tokens, while *FastV* trims visual tokens by discarding redundant patches after the second transformer layer. We implement LightThinker as a lightweight controller on top of QWEN's language blocks and adopt the original hyperparameters, using $\lambda_{\text{entropy}}{=}5\text{e}^{-3}$. FastV is inserted into the frozen CLIP encoder using its default layer-2 cutoff rule. To ensure a fair comparison under similar adaptation budgets, we jointly fine-tune the two modules for one epoch on the VoCoT corpus.

**Heima** (Shen et al., 2025a) performs reasoning in a compressed latent "hidden thinking" representation by mapping intermediate thoughts into a single latent token that is only decoded at the end. To adapt it to multimodal inputs, we feed interleaved image–text features to the Heima encoder so that the latent vector can capture fused cross-modal cues. Concretely, each intermediate step is represented as a 128-dimensional latent, while the decoder remains frozen and is used only for reconstructing the final output when required. We follow the official setup and fine-tune the encoder only, using AdamW ($\beta_1{=}0.9$, $\beta_2{=}0.95$, learning rate $1{\times}10^{-4}$) for 3 epochs on the VoCoT corpus.

**CODI** (Shen et al., 2025b) distills explicit natural-language chain-of-thought reasoning into a continuous latent space via teacher–student learning. An explicit-CoT teacher model is first trained with supervised reasoning traces, while a student model learns to perform implicit reasoning by aligning its hidden states with those of a designated teacher token. This alignment encourages the student to internalize reasoning steps without generating intermediate text. We adapt CODI to multimodal spatial reasoning by conditioning both teacher and student on interleaved image–text inputs, enabling the latent representations to capture fused cross-modal semantics. Following the official setup, we train the student model using hidden-state distillation with a frozen teacher.

**PCCoT** (Wu et al., 2025b) accelerates continuous chain-of-thought reasoning by updating latent thought tokens in parallel rather than sequentially, inspired by Jacobi-style iterative refinement. Instead of autoregressively generating reasoning

steps, PCCoT maintains a fixed set of latent reasoning tokens that are iteratively refined across multiple parallel iterations, improving both training and inference efficiency. We extend PCCoT to the multimodal setting by initializing latent tokens from interleaved image and text features, allowing the parallel updates to operate over fused multimodal representations. We follow the official implementation, using the same iteration schedule and optimization settings.

**SparseVLM** (Zhang et al., 2025d) reduces vision-language compute by selecting a sparse subset of image tokens before they enter the language backbone. We integrate SparseVLM into the QWEN stack by applying its learned token selector on CLIP-derived visual features, using the target sparsity level $\rho$. The selector ranks patches by predicted importance and filters out low-saliency/background tokens while keeping object-relevant cues. Following the official procedure, we fine-tune the pruning module together with QWEN for one epoch on the VoCoT corpus, using AdamW with $\beta_1{=}0.9$, $\beta_2{=}0.95$ and learning rate $1{\times}10^{-4}$.

**DiffThinker** (He et al., 2025) is a generative multimodal spatial reasoning framework that reformulates vision-centric reasoning as a native image-to-image generation process using diffusion models, rather than autoregressive token decoding. Following the official setup, DiffThinker is built upon QWEN-IMAGE-EDIT and implemented using a multimodal diffusion transformer (MMDiT) trained with flow matching in the latent space of a VAE. Given a visual input and textual instruction, the model directly generates a solution image that encodes the complete reasoning trajectory in visual space, which is subsequently parsed into symbolic outputs for evaluation. Training is performed task-specifically using supervised flow matching, where the velocity field is learned via mean squared error between predicted and target flows under identical data splits as the QWEN baselines. At inference time, reasoning is executed by solving a fixed-step ordinary differential equation using an Euler solver, yielding deterministic inference costs independent of reasoning depth.

## C. Additional Analysis

### C.1. Why Low-Frequency Spectral Restriction Lowers the Lipschitz Constant

We consider flow-matching dynamics parameterized by a velocity field

$$\frac{dx(t)}{dt} \;=\; v_\theta(x(t), t, h), \tag{13}$$

and study Lipschitz smoothness w.r.t. the state $x$ (fixing $(t, h)$). We show that restricting the state evolution to a low-frequency cosine subspace yields a vector field whose Lipschitz constant cannot increase, and typically decreases strictly when the largest Jacobian gain involves high-frequency directions.

**Spectral restriction.** Let $C \in \mathbb{R}^{d \times d}$ be an orthogonal block cosine projection matrix:

$$C^\top C \;=\; I, \tag{14}$$

and let $P \in \mathbb{R}^{m \times d}$ select the $m$ low-frequency cosine coefficients. Define the low-frequency coordinate $z \in \mathbb{R}^m$ and the corresponding embedded state $x \in \mathbb{R}^d$ by

$$z \;=\; PCx, \qquad x \;=\; C^\top P^\top z, \tag{15}$$

and denote the projector onto the active (low-frequency) subspace in cosine coordinates by

$$\Pi \;=\; P^\top P. \tag{16}$$

We define the restricted vector field in the active coordinates as

$$g(z, t, h) \;=\; PC\, v_\theta(C^\top P^\top z, t, h), \tag{17}$$

so the restricted dynamics is $dz(t)/dt = g(z(t), t, h)$.

**Lemma C.1** (Projection cannot increase Lipschitzness). *Assume that for each fixed $(t, h)$, $v_\theta(\cdot, t, h)$ is L-Lipschitz:*

$$\left\| v_\theta(x, t, h) - v_\theta(x', t, h) \right\|_2 \;\le\; L \left\| x - x' \right\|_2, \qquad \forall x, x' \in \mathbb{R}^d. \tag{18}$$

*Then $g(\cdot, t, h)$ defined in (17) is L-Lipschitz in z, and hence*

$$L_{\mathrm{spec}} \;\le\; L. \tag{19}$$

*Proof.* Fix $(t, h)$. For any $z, z' \in \mathbb{R}^m$, set $x = C^\top P^\top z$ and $x' = C^\top P^\top z'$. Using (17) and operator norms,

$$\|g(z, t, h) - g(z', t, h)\|_2 \ \leq \ \|PC\|_2 \, \|v_\theta(x, t, h) - v_\theta(x', t, h)\|_2. \tag{20}$$

Since $C$ is orthogonal and $P$ is a coordinate selection, $\|PC\|_2 \leq 1$. Applying (18) gives

$$\|g(z, t, h) - g(z', t, h)\|_2 \ \leq \ L \, \|x - x'\|_2. \tag{21}$$

Finally, $x - x' = C^\top P^\top (z - z')$ and the embedding $P^\top$ is an isometry from $\mathbb{R}^m$ to the selected subspace, so

$$\|x - x'\|_2 = \|C^\top P^\top (z - z')\|_2 = \|P^\top (z - z')\|_2 = \|z - z'\|_2. \tag{22}$$

Combining (21)–(22) yields $\|g(z, t, h) - g(z', t, h)\|_2 \leq L\|z - z'\|_2$, proving the claim. $\qquad \square$

**Theorem C.2** (Jacobian block controls the restricted Lipschitz constant). *Assume $v_\theta(\cdot, t, h)$ is differentiable in $x$ and let*

$$J(x, t, h) \ = \ \nabla_x v_\theta(x, t, h), \tag{23}$$

*with cosine-basis Jacobian*

$$\tilde{J}(x, t, h) \ = \ C \, J(x, t, h) \, C^\top. \tag{24}$$

*Then the Jacobian of $g$ satisfies*

$$\nabla_z g(z, t, h) \ = \ P \, \tilde{J}(C^\top P^\top z, t, h) \, P^\top, \tag{25}$$

*and therefore*

$$L_{\text{spec}}(t, h) \ \triangleq \ \sup_z \big\| \nabla_z g(z, t, h) \big\|_2 \ = \ \sup_x \big\| P \, \tilde{J}(x, t, h) \, P^\top \big\|_2 \ \leq \ \sup_x \big\| \tilde{J}(x, t, h) \big\|_2. \tag{26}$$

*In particular, $L_{\text{spec}}(t, h) \leq L(t, h)$ where $L(t, h) \triangleq \sup_x \|J(x, t, h)\|_2$. Moreover, if*

$$\sup_x \big\| P \, \tilde{J}(x, t, h) \, P^\top \big\|_2 \ < \ \sup_x \big\| \tilde{J}(x, t, h) \big\|_2, \tag{27}$$

*then $L_{\text{spec}}(t, h) < L(t, h)$ (a strict reduction).*

*Proof.* Equation (25) follows from the chain rule applied to (17), using that $PC$ and $C^\top P^\top$ are constant matrices. The identity in (26) follows by taking operator norms and suprema. The inequality in (26) uses $\|PAP^\top\|_2 \leq \|A\|_2$ for any matrix $A$ (since $P$ and $P^\top$ have operator norm 1). Finally, the strict claim is immediate from (27). $\qquad \square$

**Corollary C.3** (Implication for solver step size). *For an explicit Euler step $\Phi_{\Delta t}(x) = x + \Delta t \, v_\theta(x, t, h)$, its Lipschitz factor is bounded by $1 + \Delta t \, L(t, h)$. For the restricted dynamics, the corresponding bound is $1 + \Delta t \, L_{\text{spec}}(t, h)$. Hence, whenever $L_{\text{spec}}(t, h) < L(t, h)$, the same Lipschitz-based stability criterion permits a larger $\Delta t$, reducing the number of solver steps at inference.*

*Proof.* For any $x, x'$,

$$\|\Phi_{\Delta t}(x) - \Phi_{\Delta t}(x')\|_2 \leq \|x - x'\|_2 + \Delta t \, \|v_\theta(x, t, h) - v_\theta(x', t, h)\|_2 \leq \big(1 + \Delta t \, L(t, h)\big) \|x - x'\|_2. \tag{28}$$

The restricted case follows by replacing $v_\theta$ with $g$ and $L(t, h)$ with $L_{\text{spec}}(t, h)$. $\qquad \square$

Lemma C.1 shows that orthogonal cosine projection and low-frequency restriction are non-expansive, so the Lipschitz constant cannot increase. Theorem C.2 refines this by identifying the restricted Jacobian as the low-frequency block $P\tilde{J}P^\top$; a strict reduction arises whenever the maximal Jacobian gain of $\tilde{J}$ cannot be realized within the low-frequency subspace (27). This explains why low-frequency spectral restriction typically yields smoother ODE dynamics and supports larger solver steps (Corollary C.3) in deterministic inference.

---

**Algorithm 1** *SpecFlow* multi-hop inference with spectral-progressive cosine-space flow matching and CFG

---

**Require:** Initial visual input $x_{\text{vis}}$, initial textual query $x_{\text{txt}}$, hops $H$
**Require:** Block Cosine Projection / inverse: $D_b(\cdot)$, $D_b^{-1}(\cdot)$; mask schedule $M(t) \in \{0, 1\}^{b \times b}$ for $t \in [0, 1]$
**Require:** Velocity model $u_\theta(\cdot, t, c)$ supporting conditional $c$ and unconditional $\varnothing$
**Require:** Guidance scale $w > 0$, Euler steps $T$ (fixed-step solver), AR text generator $G_t(\cdot)$
**Ensure:** Textual thoughts $\{\hat{t}_i\}_{i=1}^H$, visual states $\{\hat{v}_i\}_{i=1}^H$
 1: Initialize textual trace $\hat{t}_{\leq 0} \leftarrow \emptyset$
 2: Initialize visual state $\hat{v}_0 \leftarrow x_{\text{vis}}$
 3: **for** $i = 0$ to $H - 1$ **do**
 4:     Form hop conditioning $c_i \leftarrow (x_{\text{txt}}, \hat{t}_{\leq i})$ {query + accumulated text trace}
 5:     Set coefficients $X_i^{(0)} \leftarrow D_b(\hat{v}_i)$
 6:     Set step size $\Delta \leftarrow 1/T$
 7:     **for** $k = 0$ to $T - 1$ **do**
 8:         $t_k \leftarrow k/T$
 9:         $\tilde{X}^{(k)} \leftarrow M(t_k) \odot X^{(k)}$ {spectral filtering (broadcast over blocks)}
10:         $u_{\text{uncond}} \leftarrow u_\theta(\tilde{X}^{(k)}, t_k, \varnothing)$
11:         $u_{\text{cond}} \leftarrow u_\theta(\tilde{X}^{(k)}, t_k, c_i)$
12:         $u_{\text{guid}} \leftarrow u_{\text{uncond}} + w(u_{\text{cond}} - u_{\text{uncond}})$ {CFG on velocity}
13:         $X^{(k+1)} \leftarrow X^{(k)} + \Delta \, u_{\text{guid}}$ {Euler update of ODE}
14:     **end for**
15:     $\hat{v}_{i+1} \leftarrow D_b^{-1}(X^{(T)})$ {visual thought / overwritten visual workspace}
16:     $\hat{t}_{i+1} \sim G_t(x_{\text{txt}}, \hat{t}_{\leq i}, \hat{v}_{i+1})$ {AR text conditioned on current visual state}
17: **end for**

---

## C.2. Low-Frequency Stabilization and Progressive Spectral Refinement

A key design choice in *SpecFlow* is to represent intermediate visual thoughts in the cosine domain, where low-frequency coefficients capture the global spatial structure most relevant to reasoning, including layout, topology, connectivity, and relative geometry. Since intermediate visual states mainly serve as reasoning scaffolds rather than photorealistic outputs, emphasizing low-frequency components provides a compact and stable visual workspace.

Based on this observation, *SpecFlow* adopts a spectral-progressive refinement strategy. During each flow-based update, the active spectral mask starts from low-frequency coefficients and gradually expands to include mid- and high-frequency bands when additional spatial precision is useful. This coarse-to-fine process preserves the stability of low-frequency reasoning while avoiding the information loss that may occur if the workspace is restricted to a fixed low-frequency subspace throughout the entire update.

This design balances efficiency and reasoning fidelity. Because only selected spectral coefficients are active at each integration step, flow matching operates in a smoother and more constrained space, reducing unnecessary computation compared with dense visual-state updates. Meanwhile, the expandable spectral budget allows the model to recover finer spatial cues when needed, while still preventing visual-token accumulation and maintaining predictable memory usage.

The fixed low-frequency variant is therefore an efficiency-oriented special case of *SpecFlow*. It keeps only the lowest-frequency coefficients active throughout the flow trajectory, which can be sufficient for simple spatial reasoning tasks but may discard useful mid-frequency cues in more complex settings. Thus, our default design uses spectral-progressive refinement, while the fixed low-frequency setting serves as an ablation illustrating the trade-off between minimal computation and spatial fidelity.

## D. Algorithm

Algorithm 1 summarizes the multi-hop inference procedure of *SpecFlow*. At hop $i$, we first construct the hop-level condition $c_i$ by concatenating the original textual query $x_{\text{txt}}$ with the accumulated textual trace $\hat{t}_{\leq i}$. We then initialize the cosine-space visual workspace from the previous visual state by projecting it into blockwise cosine coefficients, yielding $X_i^{(0)} \leftarrow D_b(\hat{v}_i)$. To generate the hop-specific visual thought, we evolve the cosine coefficients using a fixed-step ODE solver with $T$ Euler steps. At each step $t_k = k/T$, a time-dependent spectral mask $M(t_k)$ is applied to restrict updates to the active cosine

---

**Algorithm 2** Training $u_\theta$ by cosine-space flow matching with spectral masking and condition dropout

---

**Require:** Data sample $x_0$ (image / visual state), condition $c$ (e.g., hop text), dropout prob. $p_{\text{drop}}$
**Require:** Block Cosine Projection: $D_b(\cdot)$; mask schedule $M(t)$; velocity model $u_\theta$
**Ensure:** Updated parameters $\theta$
 1: Sample noise $x_1 \sim \mathcal{N}(0, I)$
 2: Transform endpoints: $X_0 \leftarrow D_b(x_0), \quad X_1 \leftarrow D_b(x_1)$
 3: Sample $t \sim \mathcal{U}(0, 1)$
 4: Interpolate in coefficient space: $X_t \leftarrow (1-t)X_1 + tX_0$
 5: Target velocity: $\dot{X}_t \leftarrow X_0 - X_1$
 6: With prob. $p_{\text{drop}}$, set conditioning $\tilde{c} \leftarrow \varnothing$ else $\tilde{c} \leftarrow c$
 7: Predict: $\hat{u} \leftarrow u_\theta\big(M(t) \odot X_t, \ t, \ \tilde{c}\big)$
 8: Compute loss: $\mathcal{L}_{\text{FM}} \leftarrow \|\hat{u} - \dot{X}_t\|_2^2$
 9: Update parameters $\theta$ using $\nabla_\theta \mathcal{L}_{\text{FM}}$

---

subspace, producing $\tilde{X}^{(k)} = M(t_k) \odot X^{(k)}$. The masked coefficients are then processed by the learned velocity field $u_\theta(\cdot, t, c)$ under both unconditional and conditional contexts. We combine these velocities using classifier-free guidance (CFG) in velocity space to control cross-modal alignment, as defined in Equation 9. We then perform an Euler update of the cosine coefficients,

$$X^{(k+1)} \ = \ X^{(k)} + \Delta\, u_\theta^{\text{guid}}\Big(X^{(k)}, t_k; c_i\Big), \qquad \Delta = 1/T, \tag{29}$$

and repeat this update for $k = 0, \ldots, T-1$. After completing $T$ steps, we invert the block cosine projection to obtain the hop visual state $\hat{v}_{i+1} = D_b^{-1}(X^{(T)})$. Finally, the next textual thought $\hat{t}_{i+1}$ is generated autoregressively by conditioning the text generator on the query, the accumulated textual trace, and the current visual state, i.e., $\hat{t}_{i+1} \sim G_t(x_{\text{txt}}, \hat{t}_{\leq i}, \hat{v}_{i+1})$. Because the visual workspace is overwritten at each hop, both memory usage and computation remain bounded regardless of reasoning depth.

Algorithm 2 describes how the cosine-space velocity field $u_\theta$ is trained using flow matching with condition dropout. Given a target visual sample $x_0$, we first sample a Gaussian noise image $x_1 \sim \mathcal{N}(0, I)$ and project both endpoints into blockwise cosine coefficients, $(X_0, X_1) = (D_b(x_0), D_b(x_1))$. We then sample a continuous time variable $t \sim \mathcal{U}(0, 1)$ and linearly interpolate the coefficients as $X_t = (1-t)X_1 + tX_0$. Under this interpolation, the target velocity remains constant and equals $\dot{X}_t = X_0 - X_1$. The same spectral mask $M(t)$ is applied to the model input, and conditioning is randomly dropped with probability $p_{\text{drop}}$ by setting $\tilde{c} \leftarrow \varnothing$, enabling CFG during inference. The model is trained to regress the target velocity using the standard flow-matching objective defined in Equation 8. Model parameters $\theta$ are then updated by backpropagating the gradient $\nabla_\theta \mathcal{L}_{\text{FM}}$. Together, these two algorithms enable hop-wise cosine-space visual thought generation guided by text while maintaining bounded visual state size and stable efficiency across long reasoning horizons.

## E. Additional Experimental Results

### E.1. Impact of block size on reasoning quality

Figure 7 studies how the block size $b$ in block-wise cosine projection trades off reasoning quality against memory consumption. When $b$ increases from 4 to 16, the KV cache consistently decreases on both Winoground and FrozenLake, indicating that coarser block partitioning reduces the effective token granularity and thus lowers attention-state storage. However, this memory reduction comes with a clear accuracy drop, since larger blocks mix spatial content over wider regions and weaken the locality needed to preserve fine-grained spatial relations during intermediate thought updates. In contrast, very small blocks better retain local structure and improve accuracy, but they increase the cache footprint and erode the memory advantages of our design. Across both tasks, $b{=}8$ achieves the best balance, which is consistent with the long-standing JPEG heuristic and suggests that an $8{\times}8$ cosine block provides sufficient locality for spatial reasoning while keeping the KV cache modest.

### E.2. Prompt and visualization of Maze

**Maze prompt template.** We use a lightweight, hop-wise controller prompt that (i) parses the current Maze observation, (ii) produces a short textual guidance string to steer *SpecFlow*'s flow-based visual-thought update via classifier-free guidance (CFG), and (iii) emits an aggregated action plan once the route is determined. Concretely, at hop $i$ the controller outputs

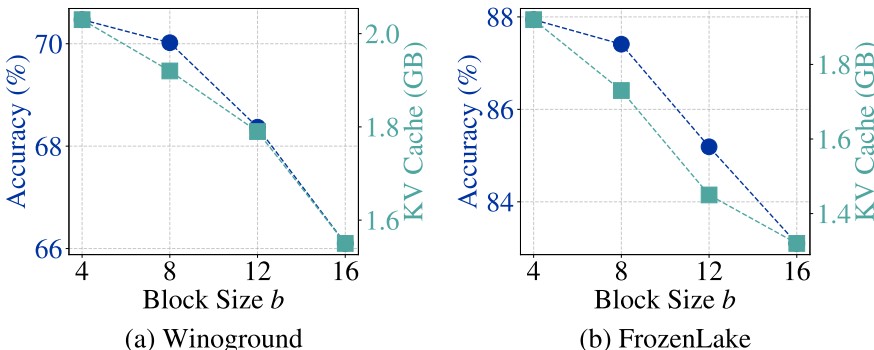

(a) Winoground

(b) FrozenLake

*Figure 7.* **Effect of spectral block size** $b$ **on reasoning accuracy and KV-cache usage.** We vary the block size used for block-wise cosine projection and report task accuracy (left y-axis) together with the total KV cache footprint (right y-axis) on (a) Winoground and (b) FrozenLake. Smaller blocks preserve finer local structure and yield higher accuracy, but increase the KV cache due to higher token granularity. Larger blocks reduce the cache footprint, but introduce coarser spectral aggregation that degrades reasoning quality. The standard JPEG choice $b=8$ provides a strong accuracy–memory trade-off across both tasks.

a one-sentence guidance `GUIDE` and a short action prefix `A_PREFIX` of 1–6 moves, and the model updates a compact visual workspace by running a small number of ODE steps under this guidance. The procedure repeats until the controller terminates with a single `ACTION_PLAN` string, which is used as the final answer. This design ensures that only the current workspace and a short textual trace need to be kept in context, avoiding the unbounded growth of dense multimodal tokens.

---

**System:** You are a professional Maze solver. Your goal is to reach the target without stepping into obstacles.
**Inputs:** (1) Current Maze image. (2) Allowed actions: {L,R,U,D}.
**Task:** Solve the Maze by planning a safe path from the start to the goal.

***SpecFlow* interface:** At hop $i$, output a short guidance text `GUIDE` that will be used as classifier-free guidance to update a compact visual workspace. You will then receive the updated workspace image for the next hop.

**Instructions:** (1) Identify the start, goal, obstacles, and free regions from the current image/workspace.
(2) Choose a short subgoal that moves closer to the goal while avoiding obstacles.
(3) Output `GUIDE` as one concise sentence describing only the subgoal and safety constraints.
(4) Output `A_PREFIX` as the next 1–6 actions consistent with `GUIDE`.
(5) When the full route is determined, output `ACTION_PLAN` as a single concatenated action string and stop.

**Output format (return exactly one of the following):**
```
GUIDE: <one sentence>
A_PREFIX: <actions>
ACTION_PLAN: <concatenated actions>
```

---

**Visualization protocol.** To visualize multi-hop reasoning, we render each intermediate visual thought (workspace) as an overlay on the Maze and draw the partial trajectory accumulated up to hop $i$ as a red polyline. After each hop, we decode the current workspace into a grid-aligned map, overlay the current planned segment, and append it to the previously drawn prefix to form the trajectory shown in the next panel. The last panel corresponds to the complete plan represented by `ACTION_PLAN`. This visualization directly reflects the key property of *SpecFlow*: intermediate spatial states are updated by overwriting a bounded workspace, which enables long-horizon planning without appending a growing sequence of intermediate images into the autoregressive context.

### E.3. MiniBehavior: prompt and visualization

**MiniBehavior prompt template.** Different from Maze, MiniBehavior requires executing a grounded *object-manipulation* plan with preconditions, where the agent must first *fetch* a target object and then *place* it at a designated receptacle. We therefore use a hop-wise controller prompt that (i) parses the current grid observation and the inventory state, (ii) outputs a concise guidance sentence to steer *SpecFlow*'s visual-workspace update via CFG, and (iii) emits an aggregated action plan once the full sequence is determined. In our running example, the goal is to fetch the `printer` and place it onto the `table`. We visualize the intermediate plan as a red rectangle that marks the current subgoal region, which first locks onto the printer location and then switches to the table location after pickup.

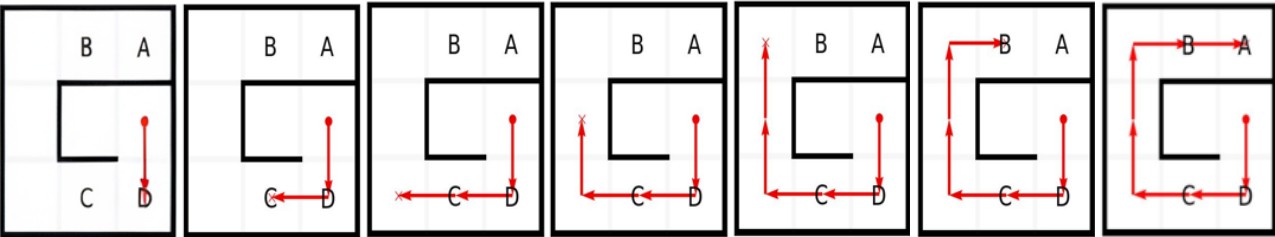

*Figure 8.* **Success case of multi-hop Maze planning with *SpecFlow*.** Each panel shows the intermediate visual workspace at one hop, overlaid with the current planned trajectory in red. The route is progressively extended while preserving global Maze geometry, and the final hop yields a coherent collision-free path that reaches the goal.

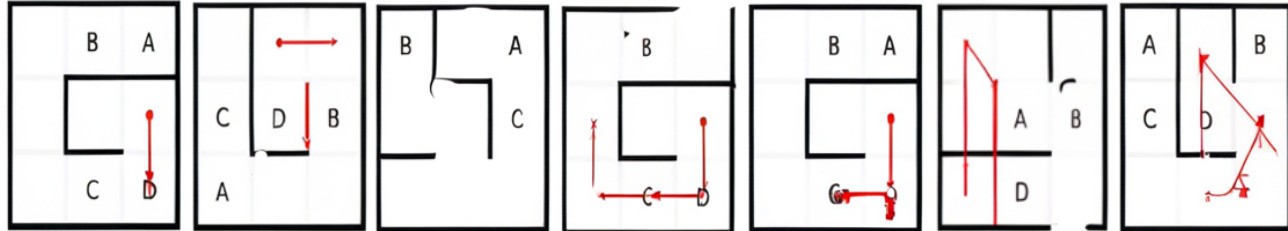

*Figure 9.* **Failure case under insufficient text alignment (under-aligned CFG).** In *SpecFlow*, the CFG scale $w$ controls how strongly textual guidance constrains the flow update of the visual workspace. When the model is under-aligned, the workspace update is not sufficiently anchored by the intended route and constraints, and the generator may drift toward visually plausible but incorrect structures, producing spurious walls/openings or inconsistent path segments ("artificial facts"). This suggests that Maze planning requires adequate alignment to keep the workspace faithful to the intended constraints, motivating careful tuning of $w$ and hop-wise guidance scheduling.

---

**System:** You are a professional MiniBehavior agent. You must complete the task by moving and interacting safely and efficiently.
**Inputs:** (1) Current MiniBehavior grid image. (2) Allowed actions: {Left, Right, Up, Down, Pickup, Drop, Toggle}. (3) Current inventory (empty or holding an object).
**Task:** Fetch the `printer`, then place the `printer` on the `table`.

***SpecFlow* interface:** At hop $i$, output a short guidance text `GUIDE` used as classifier-free guidance to update a compact visual workspace. You will then receive the updated workspace image for the next hop.

**Instructions:** (1) Identify the agent position, the `printer`, the `table`, obstacles, and free cells from the current image/workspace.
(2) Determine the current stage based on inventory: if not holding the printer, the next subgoal is *reach and pick up the printer*; otherwise the next subgoal is *reach and place the printer on the table*.
(3) Output `GUIDE` as one concise sentence that specifies the next subgoal and necessary interaction (`Pickup` or `Drop`), and includes safety constraints (avoid obstacles, stay in bounds).
(4) Output `A_PREFIX` as the next 1–7 actions consistent with `GUIDE`.
(5) When the entire plan is determined, output `ACTION_PLAN` as a single concatenated action sequence using the tokens: `L,R,U,D,PICK,DROP,TOG`.

**Output format (return exactly one of the following):**
```
GUIDE: <one sentence>
A_PREFIX: <tokens separated by commas>
ACTION_PLAN: <tokens separated by commas>
```

---

**Visualization protocol.** We render each intermediate *SpecFlow* workspace as an overlay on the grid observation and highlight the current subgoal with a red rectangle. In the fetch stage, the red rectangle focuses on the printer region, indicating that the workspace has localized the target object and is planning a collision-free route toward it. After pickup, the subgoal switches and the red rectangle moves to the table region, reflecting the transition to the placement stage. This two-stage evolution provides a direct qualitative signal that *SpecFlow* maintains a bounded workspace while supporting structured, long-horizon manipulation plans.

### E.4. FrozenLake: prompt and visualization

**FrozenLake prompt template.** We adopt the same hop-wise controller design as in the Maze and MiniBehavior settings, but specialize the instructions to FrozenLake, where the key constraint is to avoid slipping into holes while reaching the goal. At each hop $i$, the controller parses the current grid observation, produces a concise guidance sentence `GUIDE` to

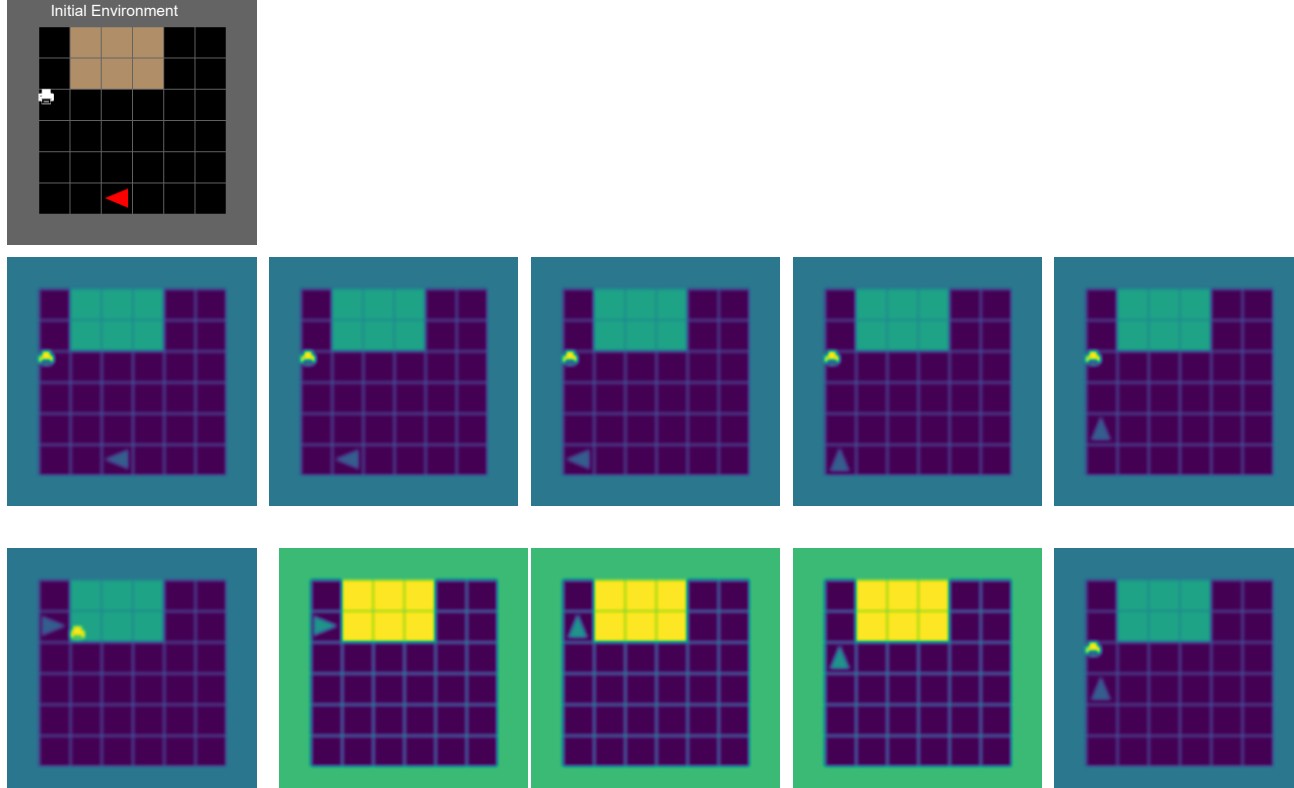

*Figure 10.* **Qualitative MiniBehavior example: fetch then place.** We visualize multi-hop visual thoughts for a two-stage manipulation task. The red rectangle denotes the current subgoal region encoded in the workspace, which first targets the `printer` for pickup and then switches to the `table` for placement. The sequence illustrates how *SpecFlow* overwrites a compact workspace to track subgoals and progress, enabling multi-step execution without accumulating dense intermediate visual tokens in the autoregressive context.

steer *SpecFlow*'s flow-based workspace update via CFG, and emits a short action prefix `A_PREFIX`. The process terminates once the controller is confident about the full route and outputs an aggregated `ACTION_PLAN`. This prompt structure aligns naturally with *SpecFlow*'s bounded-workspace reasoning, since the controller only conditions on the current observation and the current workspace snapshot.

---

**System:** You are a professional FrozenLake solver. Your goal is to reach the gift (goal) without stepping onto ice holes.
**Inputs:** (1) Current FrozenLake grid image. (2) Allowed actions: {L,R,U,D}.
**Safety rule:** Any move onto a hole is an immediate failure. Moves outside the grid are invalid.
**Task:** Output a safe action plan from the start to the goal.

***SpecFlow* interface:** At hop $i$, output `GUIDE` as classifier-free guidance to update a compact visual workspace. You will then receive the updated workspace image for the next hop.

**Instructions:** (1) Identify start, goal, holes, and safe cells from the current image/workspace.
(2) Propose a short subgoal that reduces distance to the goal while avoiding holes.
(3) Output `GUIDE` as one sentence that states the subgoal and the constraint "avoid holes".
(4) Output `A_PREFIX` as the next 1–6 actions consistent with `GUIDE`.
(5) When the full route is determined, output `ACTION_PLAN` as a single concatenated action string and stop.

**Output format (return exactly one of the following):**
```
GUIDE: <one sentence>
A_PREFIX: <actions>
ACTION_PLAN: <concatenated actions>
```

---

**Visualization protocol.** Figure 11 visualizes the intermediate visual thoughts produced by *SpecFlow* for a FrozenLake instance. We show the initial environment at the top-left, followed by a sequence of decoded workspaces across hops. Each workspace is rendered as a dense heatmap-like visualization that preserves the spatial layout while highlighting task-relevant structures, including hole locations and the evolving intended trajectory. This qualitative view makes the planning process explicit and illustrates that *SpecFlow* can refine a safe path over multiple hops by repeatedly overwriting a

Initial Environment

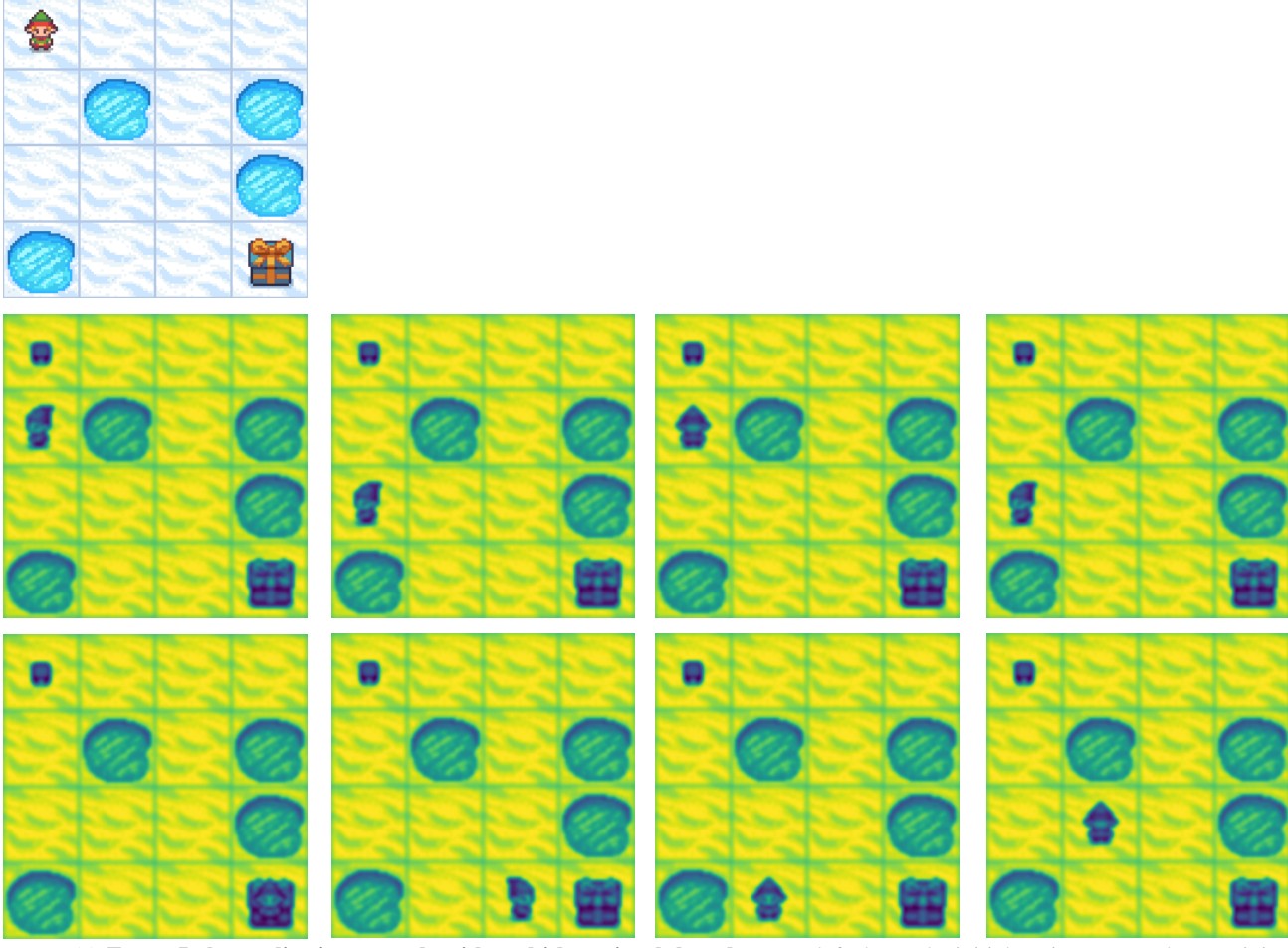

*Figure 11.* **FrozenLake qualitative example with multi-hop visual thoughts.** Top-left shows the initial environment. The remaining panels show *SpecFlow*'s decoded visual workspaces across hops under the hop-wise controller prompt. The sequence progressively refines a collision-free route that avoids holes and reaches the goal, demonstrating bounded-workspace long-horizon reasoning.

bounded workspace, rather than accumulating intermediate images in the autoregressive context.

