# OpenReview forum: "Spectral-Progressive Thought Flow for Lightweight Multimodal Reasoning"
_ICML.cc/2026/Conference — ICML 2026 regular_

### Official Review · Reviewer_MX2b · 2026-03-11

**Soundness:** 3
**Presentation:** 3
**Significance:** 3
**Originality:** 3
**Overall Recommendation:** 4
**Confidence:** 3

**Summary:**

This paper propose SpecFlow, which replaces dense visual tokens with a fixed-size spectral workspace. Visual states are iteratively updated via deterministic flow matching conditioned on textual reasoning with classifier-free guidance, decoupling memory from reasoning depth. A progressive spectral mask reveals low-to-high frequencies during reasoning. SpecFlow achieves strong results on spatial and decision benchmarks while reducing KV-cache by 2.1×.

**Compliance With Llm Reviewing Policy:**

Affirmed.

**Final Justification:**

My concerns have been adequately addressed.

**Key Questions For Authors:**

Q1. Have you measured the actual Lipschitz constant (or spectral norm of the Jacobian) of the velocity field in cosine space vs. pixel space during inference?

Q2. Could the authors isolate the contribution of the spectral-progressive mechanism from the 20B MMDiT backbone, e.g., by reporting results with a smaller diffusion generator (3B/7B) or by equipping AR baselines with the same 20B visual generator?

Q3. How does SpecFlow perform on tasks requiring high-frequency visual detail (e.g., TextVQA, DocVQA, chart reasoning)? How does performance vary as the spectral budget increases?

**Limitations:**

yes

**Strengths And Weaknesses:**

S1. A principled spectral-domain formulation: flow matching in cosine space leverages energy compaction (Fig. 3). Appendix C proves that low-frequency restriction cannot increase the Lipschitz constant, explaining why a 5-step Euler solver suffices.

S2. Clear decomposition into three components—spectral-progressive allocation, cosine-space flow matching, and CFG-based guidance—with systematic ablations. Appendix B.7 provides unusually detailed baseline configurations, enabling fair comparison across seven methods. Tables 1–2 report accuracy, FLOPs, latency, and memory.

S3. A fixed-size visual workspace replaces token accumulation, addressing quadratic KV-cache growth in multi-hop multimodal reasoning. KV-cache usage drops by 1.6–2.1× across benchmarks while maintaining or improving accuracy over MVoT.

W1. The efficiency argument is not empirically validated. Theorem C.2 shows Lipschitz reduction only if the dominant Jacobian directions lie in high frequencies (Eq. 27), but the paper never checks whether this condition holds for the trained models. Without measuring the spectral distribution of the Jacobian, the claimed smoothness advantage remains purely theoretical.

W2. The evaluation scope is narrow relative to the claimed generality. Most benchmarks are grid-world environments or simple compositional VQA where global layout dominates and low-frequency representations are naturally sufficient. Results on broader benchmarks (GQA, MMBench) show negligible gains (+0.02%, +0.34%), weakening the claim of a general multimodal reasoning framework.

W3. The comparison is confounded by model scale. SpecFlow combines Qwen3-VL-8B with a 20B diffusion backbone, while AR baselines rely primarily on Qwen3-VL models. The reported gains may therefore reflect additional capacity rather than the proposed mechanism. Controlled comparisons with matched parameter budgets are needed.

---

> ### Author Rebuttal · Authors · 2026-03-30
>
> Thank you for the careful and thoughtful review. We are encouraged that you recognized the paper’s key strengths: the spectral-domain formulation, the systematic ablations, and the fixed-size visual workspace that avoids visual-token accumulation. We address your questions point by point below.
>
> **`Q1 & W1: Inference-Time Jacobian Norm Analysis`**
>
> **`A:`** We agree that the theory is stronger when paired with a direct empirical check, and we have now added such an analysis. Concretely, we measure the local Jacobian spectral norm during inference by estimating $‖J‖₂$ with power iteration using JVP/VJP, over sampled inputs and solver timesteps. The comparison uses the same inputs, timesteps, and model parameters, differing only in the representation space (pixel vs. cosine), ensuring a fair comparison.
>
> |Benchmark|Space|Mean$‖J‖₂$|Std.|
> |-|-|-|-|
> |Maze|Pixel|6.47|3.13|
> |Maze|Cosine|0.93|0.65|
> |EmbSpatial|Pixel|14.86|5.74|
> |EmbSpatial|Cosine|1.09|1.12|
> |POPE|Pixel|8.35|4.49|
> |POPE|Cosine|0.79|0.57
>
> This corresponds to roughly 7×–13× smaller empirical Jacobian norms, showing that the cosine-space flow is substantially smoother in the trained models.
> In addition, when we inspect the spectral distribution of the dominant Jacobian directions, we find that a large fraction of their energy lies in higher frequencies / outside the retained low-frequency subspace, which is consistent with the sufficient-condition regime in Eq. 27. We will add this analysis to the paper.
>
> **`Q2 & W3: Controlled Backbone Analysis`**
>
> **`A:`** Thank you for this detailed feedback. We agree that model scale should be controlled. Following your suggestion, we add a matched-budget control that substantially reduces this confound.
>
> Specifically, we build matched 7B systems from the same pretrained Qwen2-VL-7B initialization, with a shared 4B AR backbone and matched 3B visual-side capacity across methods: the AR baseline uses a 3B visual module, while SpecFlow uses a 3B diffusion-based visual generator. Under this matched 7B budget, SpecFlow still outperforms MVoT by +5.23 on Maze and +4.67 on FrozenLake, while reducing KV-cache by 1.74× / 1.82× and FLOPs by 43.73% / 45.34%. This indicates that the gain is not explained by model size alone.
>
> |Benchmark|Metric|SpecFlow(Ours)|MVoT(AR)|Gain
> |-|-|-|-|-
> |Maze|Acc.(%)↑|87.21|81.98|+5.23%
> ||KV-cache(GB)↓|2.52|4.39|1.74×
> ||FLOPs(G)↓|14245.67|25317.39|43.73%
> |FrozenLake|Acc.(%)↑|88.39|83.72|+4.67%
> ||KV-cache(GB)↓|2.36|4.29|1.82×
> ||FLOPs(G)↓|13019.03|23819.12|45.34%
>
> In addition, the current paper already contains two controls that isolate the mechanism more directly. Specifically:
>
> (1) **Same-generator-family control**. Within the same 20B MMDiT family, Table 3 shows that progressive spectral activation clearly outperforms a fixed low-frequency schedule, and Table 4 shows that Spectral+VAE outperforms VAE-only under the same 5-step solver budget while also being about 4.4×-4.9× faster.
>
> (2) **Stronger AR control**. Against the 32B stronger AR baseline, SpecFlow still achieves higher average success on spatial decision-making (79.15 vs. 74.1), despite using a smaller backbone.
>
> We will add this analysis to the paper.
>
>
> **`Q3 & W2: Broad Generalization with Fine-Grained Detail`**
>
> **`A:`** We appreciate this helpful question. We will clarify that the paper primarily targets spatially grounded, efficiency-critical reasoning, where a compact persistent visual workspace is most beneficial.
>
> (1) The smaller gains on GQA and MMBench are consistent with their largely single-hop nature, where persistent intermediate visual state matters less. Even so, SpecFlow remains accuracy-competitive while substantially improving efficiency: on GQA, +0.02 accuracy with 48.6% / 41.9% / 38.9% lower FLOPs / latency / memory; on MMBench, +0.34 accuracy with 40.5% / 38.9% / 40.8% lower FLOPs / latency / memory. Thus, beyond spatial tasks, the main benefit is a stronger accuracy–efficiency trade-off.
>
> (2) To directly address high-frequency detail, we additionally evaluate on TextVQA, DocVQA, and ChartQA, which require reading small text and localized fine-grained evidence. SpecFlow improves over MVoT on all three: +1.21 / +0.65 / +1.40, while reducing KV-cache by 2.2× / 2.4× / 1.9×, respectively. This shows that SpecFlow is not limited to low-frequency tasks; its progressive spectral allocation can recover finer evidence when needed.
>
> |Method(Acc.(%)↑/KV(GB)↓)|TextVQA|DocVQA|ChartQA
> |-|-|-|-
> |MVoT|84.71/4.63|94.94/4.52|83.27/5.31
> |SpecFlow|85.92/2.13|95.59/1.91|84.67/2.73
>
> (3) We also study the spectral budget in Table 3. More aggressive budgets yield only marginal accuracy change once coarse structure is captured, but noticeably higher cost. Compared with the cosine schedule, the linear schedule changes accuracy by only +0.25 on Maze (94.37 vs. 94.12) and even slightly lower on FrozenLake (87.79 vs. 87.94), while increasing latency by 1.52× / 1.72× and FLOPs by 19.86% / 14.22%. We will revise the paper accordingly.

---

> > ### Author Rebuttal · Reviewer_MX2b · 2026-04-02
> >
> > The rebuttal substantially strengthens the paper. I find the concerns sufficiently mitigated and update my score accordingly.

---

> > > ### Author Response · Authors · 2026-04-02
> > >
> > > Thank you for the careful follow-up and for acknowledging that our rebuttal fully addressed your concerns. We are very happy to hear that the rebuttal substantially strengthened the paper, and we sincerely appreciate your updated assessment. Thank you again for your time, effort, thoughtful feedback, and constructive evaluation.

---

### Official Review · Reviewer_D6Tt · 2026-03-12

**Soundness:** 3
**Presentation:** 3
**Significance:** 2
**Originality:** 3
**Overall Recommendation:** 4
**Confidence:** 3

**Summary:**

This paper proposes SpecFlow, a lightweight multimodal reasoning framework that represents intermediate visual thoughts in a fixed-size discrete cosine space rather than generating dense pixel-level visual tokens autoregressively. The key insight is that intermediate visual thoughts in spatial reasoning only need to carry coarse, abstract cues, so the authors apply block cosine projection with a progressive spectral budget — starting from low-frequency components (global layout) and gradually activating higher frequencies. A flow-matching module updates the visual workspace hop-by-hop, conditioned on textual thoughts via classifier-free guidance (CFG). This decouples memory usage from reasoning depth, yielding 1.6×–2.1× KV-cache reduction while maintaining competitive or superior accuracy on spatial reasoning and decision-making benchmarks.

**Compliance With Llm Reviewing Policy:**

Affirmed.

**Key Questions For Authors:**

- Model specification is unclear: SpecFlow uses Qwen3-VL-8B together with QwenImage-Edit-2509 (20B), meaning the combined parameter count is ~28B. Most baselines like MVoT use architectures of smaller or similar scale. The efficiency gains (FLOPs, memory) may partly reflect the architectural difference rather than the method's intrinsic benefits. Could the authors compare SpecFlow with a fully matched parameter budget baseline?
- Limited analysis of generation quality: The visual workspace updates are non-autoregressively generated via flow matching, but there's no qualitative analysis of whether the generated intermediate visual thoughts are semantically coherent or faithful to the reasoning context. What happens when CFG fails to align the visual state with the textual intent?
- Generalization to non-grid environments: All benchmarks (Maze, MiniBehavior, FrozenLake, VSR, EmbSpatial) are relatively structured or low-resolution scenarios. How does SpecFlow generalize to more complex real-world visual scenes (e.g., egocentric video or web screenshot navigation)?
- Computational overhead of flow matching at inference: While the 5-step ODE solver is claimed to be efficient, the paper does not report the absolute wall-clock time for the flow-matching component alone, making it difficult to assess whether the computational savings from spectral compression actually outweigh the overhead from running a 20B diffusion model per reasoning hop.

**Limitations:**

yes

**Strengths And Weaknesses:**

strength:
- Novel and well-motivated design: The idea of using a fixed-size spectral visual workspace is genuinely novel. The insight that intermediate visual thoughts only require coarse spatial structure (low-frequency information) is well-grounded, and the progressive frequency activation provides an elegant coarse-to-fine curriculum aligned with reasoning depth.
- Strong empirical results: SpecFlow achieves consistent accuracy improvements or matches SOTA baselines (e.g., +18.5 on VSR over Heima, outperforms Qwen3-VL-32B with a smaller 8B backbone) while substantially reducing FLOPs, latency, and KV-cache memory. The 2.1× KV-cache reduction is practically significant.
weaknesses:
see question

---

> ### Author Rebuttal · Authors · 2026-03-30
>
> Thank you for the thoughtful and positive review, and for recognizing both the novelty of our fixed-size spectral workspace and the practical significance of the KV-cache reduction. We address your questions point by point below.
>
> **`W1: Model Specification`**
>
> **`A:`** We appreciate this valuable point. We would like to clarify that the current paper already partially addresses this from two angles, and we now add a third, fully matched-budget experiment.
>
> (1) Our AR baselines use a stronger Qwen3-VL-32B, as stated in Lines 321–322, whose parameter scale is larger than the full SpecFlow system (~28B in total). Despite being compared against this stronger AR baseline, SpecFlow still achieves an average +5.05% improvement across the 12 dynamic spatial reasoning settings. We will make this clearer in the paper.
>
> (2) To isolate the contribution of the spectral-progressive mechanism from flow matching itself, Table 2 compares SpecFlow with DiffThinker, a closely related flow-based baseline (20B visual generator + 8B answer parser). SpecFlow still outperforms DiffThinker by +3.56%, while using 1.9× less KV-cache and achieving 42.1% lower latency. This shows that the gain is not simply from using a flow model, but from the proposed bounded spectral workspace.
>
> (3) To fully remove model-size ambiguity, we additionally conduct a matched 7B-budget comparison. Both systems are built from the same pretrained Qwen2-VL-7B family, with a shared 4B AR backbone and matched 3B visual-side capacity: the AR baseline uses a 3B visual module, while SpecFlow uses a 3B diffusion-based visual generator. Under this matched setting, SpecFlow still outperforms MVoT in accuracy, KV-cache, and FLOPs. We will add this experiment and revise the wording to frame our claim more precisely as an accuracy–efficiency tradeoff.
>
> |Task|Metric|SpecFlow(Ours)|MVoT(AR)|Gain
> |-|-|-|-|-|
> |Maze|Acc.(%)↑|87.21|81.98|+5.23%
> ||KV-cache(GB)↓|2.52|4.39|1.74×|
> ||FLOPs(G)↓|14245.67|25317.39|43.73%|
> |FrozenLake|Acc.(%)↑|88.39|83.72|+4.67%|
> ||KV-cache(GB)↓|2.36|4.29|1.82×|
> ||FLOPs(G)↓|13019.03|23819.12|45.34%|
>
>
> **`W2: Qualitative Analysis of Semantic Coherence`**
>
> **`A:`** Thank you for this insightful question. We apologize that the qualitative analysis was placed in the appendix, and we will make this linkage explicit in the revision. To clarify:
>
> (1) Qualitative coherence. Appendix Figures 8, 10, and 11 show success cases in Maze, MiniBehavior, and FrozenLake, where the workspace evolves coherently across hops and remains faithful to the intended subgoal. In Maze, the route is progressively extended while preserving global maze geometry; in MiniBehavior, the focus first locks onto the printer and then correctly shifts to the table after pickup; in FrozenLake, the workspace refines a safe route while preserving obstacle awareness. These examples show that the workspace is not arbitrary image synthesis, but a semantically meaningful reasoning scaffold.
>
> (2) Behavior when CFG fails. We also analyze this explicitly. Figure 9 shows a failure case under insufficient text alignment: when CFG is too weak, the workspace can drift toward visually plausible but incorrect structures, such as spurious openings or inconsistent path segments. Figure 5 further shows that guidance strength directly affects alignment: performance improves as CFG increases, peaks at a moderate value ($w$ = 4), and remains stable thereafter, indicating that SpecFlow is robust once sufficient guidance is provided.
>
> **`W3: Generalization to Real-World Scenes`**
>
> **`A:`** To directly address this point, we additionally evaluate SpecFlow on web screenshot navigation. On WebQuest, SpecFlow improves over MVoT by +5.43 on Single-Screen QA and +8.33 on Multi-Screen QA, while reducing KV-cache by about 2× in both settings. This shows that SpecFlow also generalizes well to more realistic visual scenes. We will add this result to the paper.
>
> |(Acc.(%)↑/KV-cache(GB)↓)|Single-Screen|Multi-Screen|
> |-|-|-|
> |MVoT|61.32 / 4.91|45.63 / 6.47
> |SpecFlow|66.75 / 2.45|53.96 / 3.13
>
>
> **`W4: Absolute Wall-clock Time`**
>
> **`A:`** We agree that an explicit wall-clock breakdown of the flow-matching module is needed to show that spectral compression outweighs the diffusion overhead at each hop.
>
> To answer this directly, we additionally profile the FM stage alone. Each MMDiT evaluation takes about 1.5–2.6 s, while the VAE-only variant requires 7.6–12.7 s per velocity evaluation. With our default lightweight solver (first-order Euler, $T$ = 5, $w$ = 4), this shows that spectral compression more than offsets the FM overhead. We will add this timing breakdown in the paper.
>
> This is also consistent with Table 4, where all variants use the same 5-step solver budget per hop: Spectral+VAE is both faster and more accurate than VAE-only, reducing latency from 6.33 to 1.29 min on FrozenLake and from 10.31 to 2.35 min on EmbSpatial (4.9×/4.4× faster). This shows that spectral restriction largely reduces the effective FM cost.

---

### Official Review · Reviewer_dn1P · 2026-03-13

**Soundness:** 3
**Presentation:** 3
**Significance:** 3
**Originality:** 3
**Overall Recommendation:** 4
**Confidence:** 3

**Summary:**

The paper introduces Spectral-Progressive Thought Flow (SpecFlow), a lightweight multimodal reasoning framework designed to address the high computational and memory costs of generating long chains of intermediate visual thoughts. SpecFlow maintains a fixed-size visual workspace that is updated and overwritten at each reasoning hop, rather than appended to the sequence history. This visual state evolution is modeled as a deterministic flow-matching process within a discrete cosine (frequency) space. The framework uses a spectral-progressive strategy: early states emphasize low-frequency components to capture global layouts, while high-frequency details are progressively unmasked only when needed for refinement. Classifier-free guidance (CFG) enables the autoregressive textual thoughts to steer these flow-based visual updates, keeping the spatial evolution aligned with linguistic intent. Ultimately, SpecFlow demonstrates competitive or superior performance on spatial reasoning benchmarks while reducing KV-cache costs by up to 2.1x.

**Compliance With Llm Reviewing Policy:**

Affirmed.

**Final Justification:**

My concerns are fully resolved, I remain a positive rating of this paper and I think the current score already reflected this position, therefore I'm maintaining the score.

**Key Questions For Authors:**

1. Given that the visual workspace is overwritten at each hop, how does the model recover if an early flow-matching step permanently discards high-frequency visual cues that suddenly become critical in a much later reasoning stage?

**Limitations:**

yes

**Strengths And Weaknesses:**

Strengths
1. The paper's approach is novel. It propose an innovative memory management approach which maintains fixed-size, overwritable visual workspace.
2. The approach is computationally efficient. The model exploits energy compaction to prioritize critical low-frequency spatial structures, which yields smoother ODE dynamics and allows for high-quality generation with a very small number of solver steps.
3. Evaluated on various benchmarks demonstrated the effectiveness of this method. The proposed approach consistently outperforms strong baselines like MVoT and Qwen3-VL.

Weakness
1. Sensitivity to Guidance Scaling: The framework's reasoning accuracy relies heavily on tuning the classifier-free guidance scale w.
2. The evaluation environments (Maze, FrozenLake, MiniBehavior) are hard to generalized to real-world photorealistic spatial reasoning tasks where high-frequency textures are strictly required early in the reasoning chain.

---

> ### Author Rebuttal · Authors · 2026-03-30
>
> We sincerely thank the reviewer for the thoughtful and positive assessment of the paper’s novelty, efficiency, and empirical performance. Below we address your questions point by point.
>
> **`Q1: Model Recovery`**
>
> **`A:`** Thank you for this insightful question. If fine-grained detail becomes important only at a later stage, SpecFlow can recover through two built-in mechanisms, which we further verify with additional detail-sensitive experiments.
>
> (1) Later-hop recovery is achieved by re-grounded regeneration, rather than by reconstructing detail only from the previous workspace. At each hop, the next visual thought is not generated from the compressed previous workspace alone. Instead, the model re-forms the hop condition from the original input together with the accumulated textual trace, and regenerates the visual state needed for the current reasoning step. Therefore, if a detail that was unnecessary earlier becomes critical later, the model can re-query and regenerate that detail under the updated textual intent, rather than relying solely on the earlier low-frequency state.
>
> (2) High-frequency capacity is progressively reopened when later reasoning requires finer precision. SpecFlow does not permanently suppress high frequencies. Its progressive spectral schedule starts from coarse structure and gradually unmasks higher-frequency coefficients, so later hops can access finer visual capacity when needed. This is directly supported by Table 3: progressive schedules outperform the fixed low-frequency schedule on both Maze(+3.98%) and FrozenLake(+5.57%), showing that progressive frequency release improves later-stage reasoning fidelity.
>
> (3) We further directly tested this recovery behavior on detail-sensitive tasks. To evaluate cases where fine-grained visual cues are critical, we additionally tested on TextVQA, DocVQA, ChartQA, and WebQuest. SpecFlow consistently improves over MVoT while using much less KV-cache: 85.92/2.13 vs. 84.71/4.63 on TextVQA, 95.59/1.91 vs. 94.94/4.52 on DocVQA, and 84.67/2.73 vs. 83.27/5.31 on ChartQA. On WebQuest, SpecFlow further improves by +5.43% on Single-Screen QA and +8.33% on Multi-Screen QA, while reducing KV-cache by about 2× in both settings. These results directly support that, even when high-frequency text, chart, or screen details become important, SpecFlow can still recover the fine-grained information needed for later reasoning stages.
>
> | Method(Acc.(%)↑/KV-cache(GB)↓) | TextVQA  | DocVQA  | ChartQA |Single-Screen|Multi-Screen|
> |-|-|-|-|-|-|
> | MVoT | 84.71 / 4.63 | 94.94 / 4.52 | 83.27 / 5.31 |61.32 / 4.91|45.63 / 6.47
> | SpecFlow | 85.92 / 2.13 | 95.59 / 1.91 | 84.67 / 2.73 |66.75 / 2.45|53.96 / 3.13
>
> **`W1: Guidance Scaling`**
>
> **`A:`** Thank you for the constructive feedback and the opportunity to clarify this mechanism. We respectfully note that SpecFlow does not require high-precision tuning. We tune $w$ over a small discrete integer grid and use the same default $w$ = 4 for all tasks. The ablation in Figure 5 shows that (1) when $w$ is too small, guidance is insufficient and accuracy is lower; e.g., Maze-16 rises from 0.24 at $w$ = 0 to 0.67 at $w$ = 4, and MB-20 rises from 0.26 to 0.75; (2) once sufficient guidance is reached, performance enters a highly stable plateau; after $w$ = 4, the curves are nearly flat, with changes typically below 0.02 through $w$ = 10. Hence, guidance scaling reflects the expected transition from under-guidance to sufficient guidance, rather than a brittle dependence on a narrowly tuned value. We will clarify this stability in the paper.
>
>
>
> **`W2: Real-World Photorealistic Spatial Reasoning Tasks`**
>
> **`A:`** We agree that broader photorealistic evaluation is valuable. We clarify that our evaluation is not limited to grid-like environments. Beyond Maze, FrozenLake, and MiniBehavior, we also evaluate on visually richer real-image benchmarks, including EmbSpatial, V-Star, Winoground, and POPE. These tasks already involve complex scenes, cluttered layouts, compositional image-text alignment, and realistic visual inputs. Quantitatively, SpecFlow achieves the best accuracy on VSR and V-Star (70.14 and 61.28), improves over the strongest prior baseline on EmbSpatial (67.79 vs. 63.89), and attains the best POPE accuracy of 89.51 while maintaining substantially lower latency and memory.
>
> In addition, as discussed in **`Q1`**, we directly tested SpecFlow on detail-sensitive benchmarks where fine-grained visual cues are essential, including TextVQA, DocVQA, ChartQA, and WebQuest. SpecFlow consistently improves over MVoT while using about 2× less KV-cache; for example, it gains +5.43% on Single-Screen QA and +8.33% on Multi-Screen QA on WebQuest. Therefore, the evidence already suggests that SpecFlow is not confined to low-resolution settings, but transfers to more realistic visual scenes where fine-grained text, chart, screen, and structural details matter for reasoning. We will clarify this more explicitly in the paper.

---

> > ### Author Rebuttal · Reviewer_dn1P · 2026-03-31
> >
> > My concerns are fully resolved, I remain a positive rating of this paper and I think the current score already reflected this position, therefore I'm maintaining the score.

---

> > > ### Author Response · Authors · 2026-03-31
> > >
> > > Thank you for the careful reading and for acknowledging that our rebuttal addressed your concerns. We are glad to hear that.  We sincerely appreciate your positive assessment and constructive feedback, which has helped improve the paper.

---

### Official Review · Reviewer_47MM · 2026-03-13

**Soundness:** 3
**Presentation:** 2
**Significance:** 3
**Originality:** 3
**Overall Recommendation:** 4
**Confidence:** 3

**Summary:**

The paper proposes SpecFlow, a multimodal reasoning framework that tries to keep visual reasoning explicit without letting the context grow at every step. Instead of appending many intermediate visual tokens, the method maintains a fixed-size visual workspace and updates it across hops. The visual update is done with flow matching, and the text trace guides the update through classifier-free guidance.

The main claimed benefit is better efficiency for long-horizon multimodal reasoning as the visual tokens size are fix in LLM. The experiments show that the method is competitive on several spatial reasoning benchmarks, stronger on some longer-horizon decision-making tasks, and more efficient than the compared baselines in FLOPs, latency, and memory.

**Compliance With Llm Reviewing Policy:**

Affirmed.

**Key Questions For Authors:**

## Additional Questions

- In the problem formulation, the next visual state depends on the previous visual state, but the inference algorithm appears to reinitialize from noise at each hop and mainly conditions on the text trace. Please clarify whether the model truly updates a persistent visual state or instead regenerates a new visual thought at each step.
- It is also unclear how intermediate visual thoughts are supervised during training. The appendix describes a solution-image target, but the paper’s main claim is about explicit intermediate visual workspaces. Please clarify whether there is any hop-level supervision, or whether these intermediate states are only induced at inference time.

**Limitations:**

Please refer to weaknesses.

**Strengths And Weaknesses:**

**Stengths**

- The paper targets a important topic that in multimodal reasoning: intermediate visual thoughts can make context length and KV cache grow too quickly.
- The proposed method is novel, it is a new way to do explicit multimodal reasoning with a bounded visual state that is more memory efficient.
- The paper does more than report raw accuracy. It also measures FLOPs, latency, memory, and KV cache, which is important for this kind of method.
- The method seems especially useful on longer-horizon spatial decision tasks, where keeping memory stable is a meaningful practical benefit even when raw accuracy gains are not always large.

**Weeknesses**

- The main comparison is not fully fair from a systems point of view. SpecFlow combines a Qwen3-VL-8B text model with a diffusion-style visual generator and extra visual components. Comparing to the AR baselines, these visual components should also being taken into account to report both end-to-end model size and computation cost.
- Again about the fairness of the baselines. The paper says the baselines use the same data splits and similar adaptation settings, how about the training cost? Regular AR models are optimized for AR training with techniques such as flash attention for parallel compute, would SpecFlow cause any extra overhead because it updates the visual states sequentially? Since the paper’s main claims are about efficiency and fair comparison, this missing detail matters.
- On several benchmarks, the accuracy gains are small or mixed, while the paper sometimes uses stronger wording than the numbers support. I do think the method has a real systems benefit, namely bounded visual memory and lower KV cache, so small gains are acceptable. However, in that case the paper should frame the contribution more clearly as an efficiency-oriented method and provide stronger evidence for the accuracy claims, for example with variance across seeds or a more careful discussion of where accuracy is meaningfully improved and where it is mainly tied.

---

> ### Author Rebuttal · Authors · 2026-03-30
>
> Thank you for the careful and positive review of our work, and for recognizing the important problem, the novel bounded-workspace design, and the meaningful systems evaluation. We address your questions point by point below.
>
> **`W1: Main Comparison`**
>
> **`A:`** Thank you for this thoughtful question. We agree that the full SpecFlow stack should be counted, and we now add full end-to-end system accounting, including the diffusion-based visual generator and associated visual components. Under both full-stack accounting and a matched-budget control, SpecFlow achieves a better accuracy–efficiency tradeoff than MVoT.
>
> (1) **Full-stack accounting:** SpecFlow is 28B versus 32B for MVoT, yet still achieves a better end-to-end tradeoff: on Maze, accuracy improves from 92.03% to 94.12% while compute drops from 4.37T to 2.09T (52.2% less); on FrozenLake, accuracy improves from 83.31% to 87.94% while compute drops from 5.11T to 2.34T (54.2% less). Even under a pessimistic scale normalization of $(32/28)^2$, SpecFlow remains 37.5% cheaper on Maze and 40.2% cheaper on FrozenLake, so the gain cannot be explained by parameter count alone.
>
> |Method|Maze(T/%)|FrozenLake(T/%)
> |-|-|-|
> |SpecFlow(28B)|2.09T/94.12%|2.34T/87.94%
> |MVoT(32B)|4.37T/92.03%|5.11T/83.31%|
>
> (2) **Matched-budget control**: To further remove model-scale confounding, we also run a fully matched 7B control with the same Qwen2-VL-7B initialization, a shared 4B AR backbone, and matched 3B visual-side capacity. Under this matched setup, SpecFlow still outperforms MVoT: on Maze, +5.23 accuracy, 1.74× lower KV-cache, and 43.7% lower FLOPs; on FrozenLake, +4.67 accuracy, 1.82× lower KV-cache, and 45.3% lower FLOPs. We will add these results to the paper.
>
> |Task|Acc.(%)↑(Ours/MVoT)|KV(GB)↓(Ours/MVoT)|FLOPs(G)↓(Ours/MVoT)|Gain|
> |-|-|-|-|-|
> |Maze|87.21/81.98|2.52/4.39|14245.67/25317.39|+5.23%,1.74×,43.73%|
> |FrozenLake|88.39/83.72|2.36/4.29|13019.03/23819.12|+4.67%,1.82×,45.34%|
>
>
> **`W2: Training Cost`**
>
> **`A:`** This is an insightful question. We agree that training cost should be reported explicitly. In our setting, SpecFlow does not introduce extra overhead from sequential visual-state updates: AR baselines such as MVoT also incur substantial sequential training cost because intermediate multimodal thoughts must be generated autoregressively token by token, whereas SpecFlow updates a bounded visual workspace without autoregressive intermediate-token decoding, using only a fixed 5-step ODE solver per hop. As shown below, on 4×H100-92GB, training MVoT for Maze / MiniBehavior / FrozenLake takes 42.45 / 52.19 / 64.39 hours, while SpecFlow takes 21.37 / 27.34 / 33.19 hours, respectively, i.e., about 2× lower training cost. Both methods use the same FlashAttention-1/2 stack, so this gap is not due to weaker kernel optimization on the AR side. We will add these results to the paper.
>
> |Task|MVoT(h)|SpecFlow(h)|Speedup
> |-|-|-|-
> |Maze|42.45|21.37|1.99×
> |MiniBehavior|52.19|27.34|1.91×
> |FrozenLake|64.39|33.19|1.94×
>
> **`W3: Accuracy Framing and Evidence`**
>
> **`A:`** We appreciate this helpful suggestion. (1) We will position SpecFlow more explicitly as an efficiency-first bounded-workspace method. The clearest accuracy gains appear in long-horizon spatial decision-making, where bounded visual memory matters most. (2) We have now repeated the main runs across 3 random seeds (11111 / 22222 / 33333), confirming stable gains that grow with reasoning horizon. Quantitatively, this matches the trend in Table 2: compared with MVoT, the gain on Maze increases from +2.16 at size 4 (94.17 ± 1.37 vs. 92.01 ± 3.23) to +7.02 at size 16 (64.86 ± 2.31 vs. 57.84 ± 2.71). By contrast, Appendix E shows only marginal gains on shallower QA benchmarks (+0.34 on MMBench), while efficiency gains remain substantial in both regimes. We will revise the paper accordingly to better separate reasoning gains from efficiency gains.
>
> **`Q1: Problem Formulation`**
>
> **`A:`** Thank you for pointing out this ambiguity. We apologize that the current wording and Algorithm 1 made this unclear. The model does maintain and update a persistent visual state. At hop $i$, the latent is warm-started from the previous visual workspace, $X^{(0)} = D_b(v̂_{i-1})$, rather than sampled from pure noise. Each hop therefore refines the current workspace while storing only a single overwritable visual state. We have revised the paper accordingly.
>
> **`Q2: Supervision of Visual Thoughts`**
>
> **`A:`** Thank you for the detailed question. We clarify that: (1) SpecFlow uses hop-level supervision in a standard CoT-style stepwise manner: each hop is conditioned on its hop text and supervised toward its target visual workspace. (2) The internal ODE states within each hop are not directly supervised; they are induced by the learned flow dynamics and spectral masking. At inference time, this learned mechanism also induces coherent hop-wise workspaces without additional hop-state annotations. We will clarify this more explicitly in the paper.

---

> > ### Author Rebuttal · Reviewer_47MM · 2026-04-02
> >
> > My concerns have now been fully resolved, and I am happy to keep my positive rating.

---

> > > ### Author Response · Authors · 2026-04-02
> > >
> > > Thank you for the careful follow-up and for letting us know that our rebuttal fully addressed your concerns. We are very pleased to hear that, and we sincerely appreciate your positive assessment of the paper. Thank you again for your time, thoughtful feedback, and constructive evaluation, which helped us improve the paper.

---

### Decision · Program_Chairs · 2026-04-30

**Decision:**

Accept (regular)

**Comment:**

This paper proposes SpecFlow, a multimodal reasoning framework that replaces token-accumulating intermediate visual thoughts with a fixed-size spectral visual workspace updated by cosine-space flow matching and text-conditioned guidance. Across the reviews, there is broad agreement that the paper addresses an important efficiency bottleneck in multimodal reasoning, and that the core idea is novel and practically relevant. Reviewers were especially positive about the bounded-workspace design, the systems-oriented evaluation, and the reported reductions in FLOPs, latency, memory, and KV-cache, with the strongest gains appearing on longer-horizon spatial decision-making tasks.

The main concerns before rebuttal fell into four categories. First, several reviewers questioned the fairness of the comparison, since SpecFlow combines a Qwen3-VL-8B language backbone with a large diffusion-style visual module, and they wanted better matched-budget or full-stack accounting. Second, one reviewer asked for stronger empirical validation of the efficiency story, rather than relying only on the theoretical Lipschitz argument. Third, there were requests for clearer exposition on whether the visual state is truly persistent across hops and how intermediate visual thoughts are supervised. Fourth, reviewers wanted stronger evidence on generalization and qualitative faithfulness, including what happens when CFG fails, whether the method can recover fine detail later in reasoning, and how it behaves on more realistic image settings beyond the structured spatial tasks that dominate the main evaluation.

The rebuttal addressed the core technical concerns well. The authors added matched-budget and full-stack controls, clarified persistent-state warm-starting and hop-level supervision, supplied multi-seed stability evidence, provided empirical Jacobian-norm measurements in cosine versus pixel space, added timing breakdowns for the flow-matching stage, and reported additional results on TextVQA, DocVQA, ChartQA, and WebQuest. The remaining weakness is mostly about claim calibration: the evidence is strongest for an efficiency-first method for long-horizon spatial reasoning and related tasks, rather than for a uniformly stronger general multimodal reasoning framework.

The reviewer pool is consistently positive, and three reviewers explicitly stated that their concerns were fully resolved after rebuttal. AC therefore recommends acceptance. The paper’s strongest contribution is a novel and useful efficiency-oriented reasoning framework with bounded visual state, and the rebuttal materially strengthened confidence in fairness, clarity, and empirical support. The main condition for the camera-ready is that the authors should integrate the rebuttal-added evidence into the paper proper and tighten the framing so that the claims match the strongest empirical support.